# Extensive off-fault damage around the 2023 Kahramanmaraş earthquake surface ruptures

Jihong Liu [1], Sigurjón Jónsson [1] ✉, Xing Li [1], Wenqian Yao [2,3] & Yann Klinger [2]

Quantifying coseismic fault offsets for surface ruptures of major earthquakes is important for earthquake cycle and slip-rate studies, and thus for earthquake hazard assessments. However, measurements of such offsets generally underestimate fault slip due to inelastic deformation and secondary fault offsets, i.e., off-fault damage. Here, we use satellite synthetic aperture radar images to quantify off-fault damage in the two 2023 Kahramanmaraş (Türkiye) magnitude 7.8 and 7.6 earthquakes. We first derive three-dimensional coseismic surface displacements and show that on average ~35% of the coseismic slip is accommodated by off-fault damage within 5–7 km of the coseismic surface ruptures. Fault sections exhibiting geometrical complexities (e.g., bends and step-overs) experienced a higher level of off-fault damage than simpler fault sections. Our results highlight the importance of extending off-fault damage assessments to several km away from fault ruptures and indicate that fault offset measurements may underestimate slip-rate estimations by as much as a third.

Two earthquakes of magnitude Mw 7.8 and 7.6 struck within 9 h near the border between Türkiye and Syria on February 6th, 2023. These earthquakes, dominated by left-lateral strike-slip motion, occurred respectively along the southwestern part of the East Anatolian Fault (EAF) and along the Sürgü fault, which is located about 90 km north of the EAF. Overall, these faults accommodate the westward escape of the Anatolian Plate relative to the northward motion of the Arabian Plate[1–3]. The Mw 7.8 event started on a splay fault (the Narli fault, Fig. 1a), just south of the EAF, and then bilaterally ruptured the EAF, producing a ~350 km-long surface rupture[4–7]. The rupture propagation along some fault sections has been described to have progressed at super-shear rupture speeds[5,8,9]. The latter Mw 7.6 event initiated in the middle of the east–west trending Sürgü fault and caused a ~160 km-long surface rupture. While the rupture propagated eastward at sub-shear speed, super-shear rupture was documented westward[4–6]. Based on geodetic and seismological datasets,

already-published studies have modeled these two events using respectively 3–6 and 2–5 fault sections separated by bends and step-overs[4–6,8,10,11], indicating notable complexity of the rupture processes and the fault geometry.

Large strike-slip earthquakes are commonly modeled as instantaneous slip on localized planar fault planes embedded in a homogenous[12] or layered elastic half-space[13]. Over multiple earthquake cycles, the cumulative fault slip, including coseismic slip, afterslip, and possible fault creep, both at the surface and at depth should match the relative plate motion (Fig. 2)[14]. However, fault-slip inversions of strike-slip earthquakes often result in maximum slip at a depth of several km, with the slip gradually decreasing towards the surface. This has been referred to as the so-called shallow slip deficit (SSD)[15,16]. Studies based on high-resolution satellite imagery have suggested, however, that in those models the lack of slip near the surface is mostly due to the intrinsic inability of those elastic models

[1]King Abdullah University of Science and Technology (KAUST), Thuwal, Saudi Arabia. [2]Université Paris Cité, Institut de Physique du Globe de Paris, CNRS, Paris, France. [3]Institute of Surface-Earth System Science, School of Earth System Science, Tianjin University, Tianjin, China. ✉e-mail: sigurjon.jonsson@kaust.edu.sa

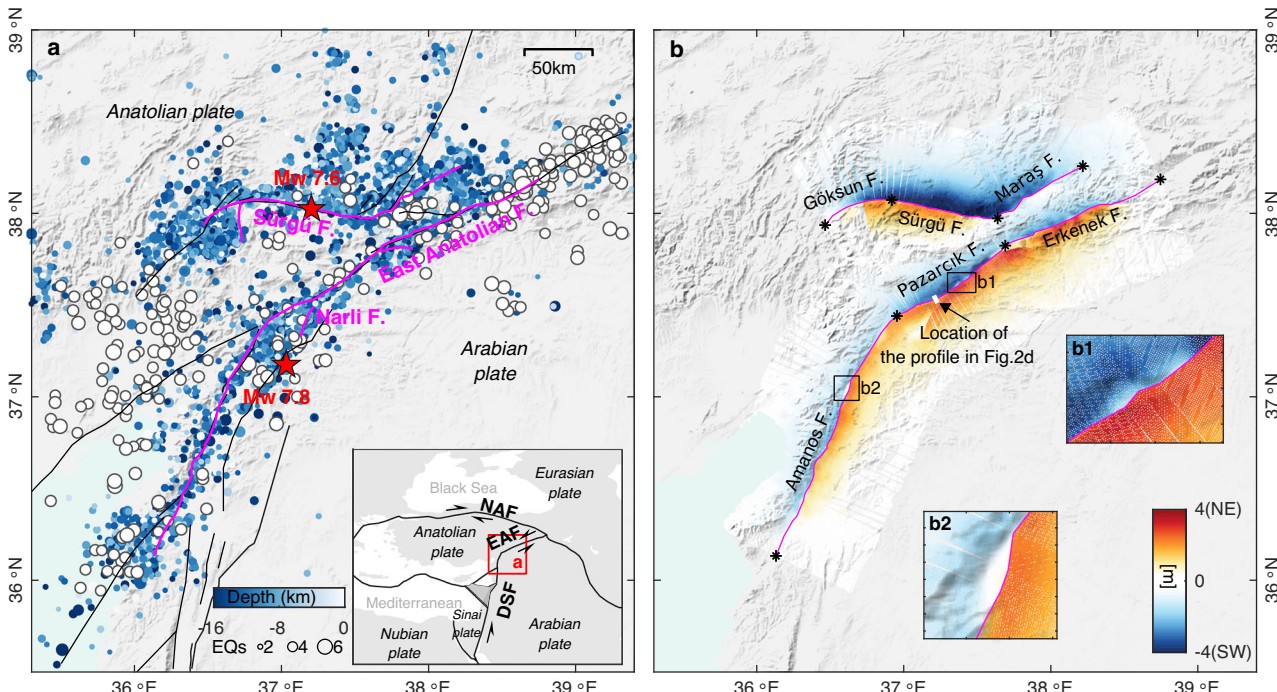

**Fig. 1 | The Kahramanmaraş earthquakes. a** Surface ruptures (magenta lines) and aftershocks[74] (colored circles) of the magnitude 7.8 and 7.6 Kahramanmaraş earthquakes of 6 February 2023. The mainshock epicenter locations (red stars), main faults (black lines), and USGS-documented earthquakes from 1900 (white circles) are shown. The inset shows the main tectonic plate boundaries in the study region[75] and the location of the Kahramanmaraş earthquakes. EAF east Anatolian fault, NAF north Anatolian fault, DSF Dead Sea fault. **b** Fault-parallel displacements derived by projecting the east–west and north–south displacements

(Supplementary Fig. 1) onto profiles (100 km in length) perpendicular to the local fault strike direction, every 0.1 km along the two fault traces. Positive and negative displacements represent northeastward and southwestward movements, respectively. Asterisks along main ruptures represent fault section boundaries. Insets b1 and b2 show zoom-in views of the on-fault rectangles illuminating examples of displacement decrease near the fault ruptures. The shaded relief background map was derived from the shuttle radar topography mission (SRTM) 3-arc seconds data[76].

to incorporate shallow off-fault deformation[14,15], likely inelastic, that can spread hundreds of meters to 1–2 km away from earthquake surface ruptures[15–19]. Both a true reduction in near-surface fault slip (i.e., SSD) and the inelastic off-fault deformation would yield a comparable observable surface displacement pattern, characterized by a decrease in on-fault slip. Here we refer to this observable phenomenon as absent surface displacement (ASD). Given that seismogenic faults are often surrounded by off-fault damage zones indicating inelastic rock deformation (e.g., warping, rigid-block rotation, microscale brittle deformations, and granular flow)[20–22], the ASD provides important clues to the extent and magnitude of off-fault damage, thus providing insights into dynamic rupture processes and earthquake hazards[23–25].

Most previous studies on off-fault damage used high-resolution optical images[17,19,26,27] and focused on near-fault regions[28–30]. The Kahramanmaraş earthquakes provide an opportunity to study off-fault damage beyond the near-fault region, as the multi-meter surface fault offsets and a range of fault geometrical complexities yielded extensive and complex off-fault deformation.

In this work, we quantify the off-fault damage both near and several km away from the main ruptures of the Kahramanmaraş earthquakes, based on complete mapping of the coseismic three-dimensional (3D) surface displacements from differential interferometry and pixel tracking of satellite synthetic aperture radar (SAR) images. The results indicate that off-fault damage consumes on average about 35% of the on-fault coseismic slip at depth. The average width of the off-fault damage zone is ~5 km, significantly wider compared with previous studies reporting damage widths of only a few hundred meters. These findings call for a reconsideration of slip-rate studies and seismic hazard assessments along major faults.

## Results
### Coseismic 3D surface displacements
We estimated the full 3D coseismic surface displacements using a strain model and variance component estimation (SM-VCE) approach[31,32] on Sentinel-1 and Advanced Land Observing Satellite-2 (ALOS-2) SAR images. We collected three tracks of Sentinel-1 images and five tracks of ALOS-2 images (Supplementary Fig. 4) and produced a total of 23 independent displacement observations (Supplementary Fig. 5) that were combined for the final 3D displacements (Supplementary Fig. 1). The advantage of using the SM-VCE method lies in that the spatial correlation between adjacent points is used to increase the signal-to-noise ratio of the target point and that the variance component estimation algorithm accurately determines the weight of observations in a posteriori and iterative way. The root mean square error of the difference between our SAR-based 3D displacements and global navigation satellite system (GNSS) observations in the area is 5.5 cm, 8.6 cm, and 5.8 cm for the east, north, and vertical displacement components, respectively (Supplementary Fig. 3). Our SAR-based east–west and north–south displacements are consistent with displacements derived from Sentinel-2 optical images[33] in both the near-fault and far-field areas (Supplementary Fig. 6). By projecting the 3D displacements back into the line-of-sight SAR geometry, we obtain residuals of the original observations (Supplementary Figs. 7 and 8). These residuals show no systematic deviations, indicating that no single input data set is leading to a bias in the 3D displacement derivation.

The resulting 3D displacement field reflects well the left lateral strike-slip mechanism with large horizontal fault-parallel displacements (Fig. 1b) and limited vertical displacements (Supplementary Fig. 1c). For the Mw 7.8 earthquake, only local uplift or subsidence

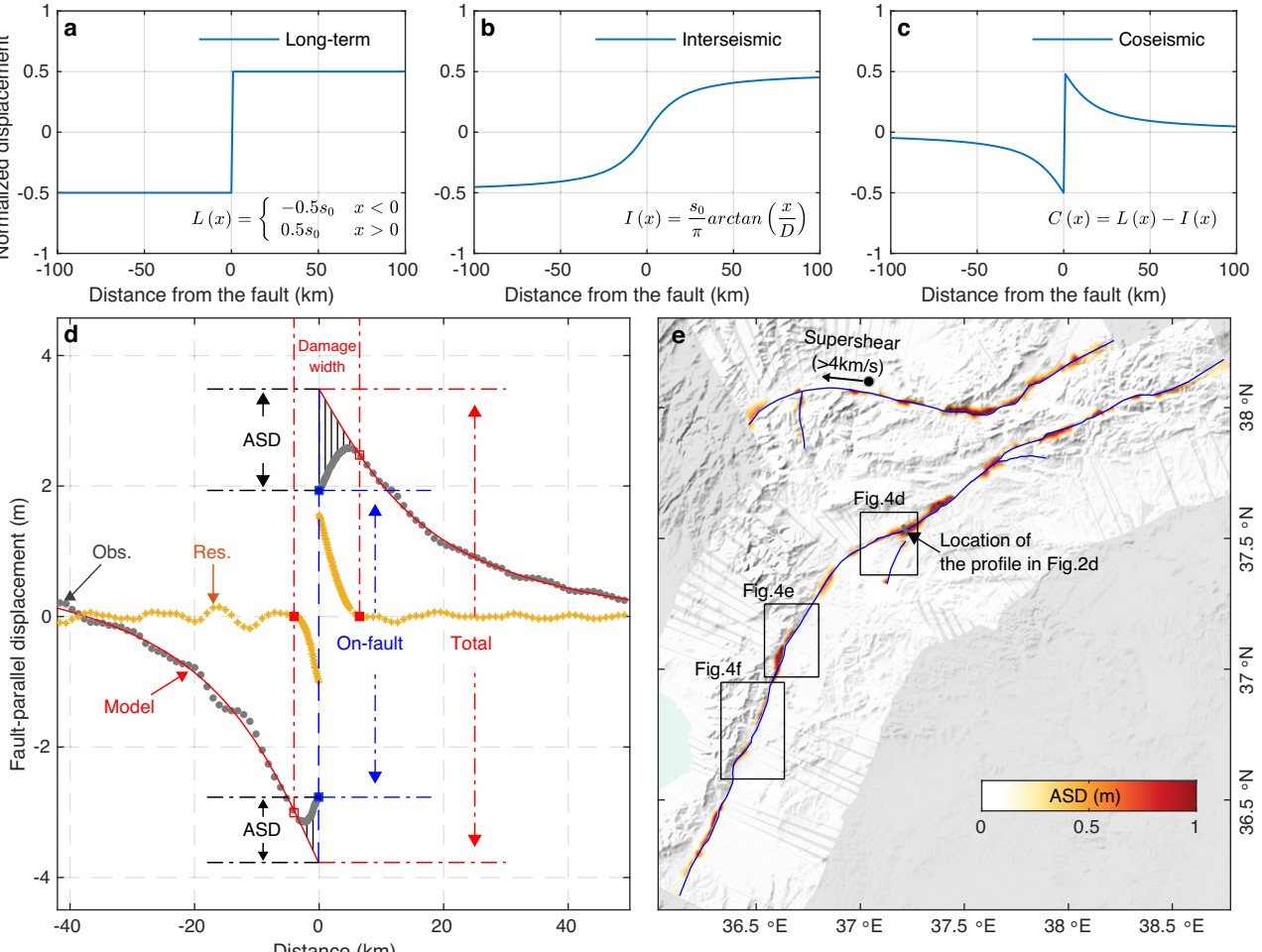

**Fig. 2 | The absent surface displacement (ASD). a–c** Fully elastic deformation across a strike-slip fault during one earthquake cycle with (**a**) the total long-term block-like motion $L(x)$ as the sum of **b** the interseismic deformation $I(x)$ (an arctan behavior) and **c** the elastic coseismic deformation $C(x)$. In **a–c** $x$ is the distance from the fault, $s_0$ slip amplitude on the fault, and $D$ locking depth during the interseismic period. **d** The quantification of ASD on a lateral profile of fault-parallel displacement observations (gray dots). The location of this profile is shown in (**e**) and Fig. 1b. The model (red curve) is obtained by fitting the observation with $C(x)$. The residual (yellow dots) of the observation with respect to the model represents the ASD, which is also depicted as vertical lines between the model (red curve) and observations (gray dots). The total deformation is defined as the difference between two peaks of the model and the on-fault offset is the observed displacement jump across the fault (at $x = 0$). By fitting the residual with a logarithmic function (see "Methods"), the extent of the ASD (red squares) and thus the damage width can be determined. In the residuals, there is a signal with a length scale of ~5 km, which can be attributed to topography-related artifacts from the pixel-offset tracking process of non-orthorectified SAR images (Supplementary Fig. 2). **e** The ASD along the main ruptures (equivalent to the absolute value of yellow dots in (**d**) within the damage width) of all fault-perpendicular profiles. The gray bar in (**e**) represents the center position of the profile (gray dots) shown in (**d**), which we picked because its displacement pattern is fairly typical of the fault-parallel displacements with ASD in both complex and simple sections along the fault (Supplementary Movie 1). The uncertainty of the profile displacements is 5–9 cm, as indicated by the comparison of the SAR-based 3D displacements with GNSS observations (Supplementary Fig. 3). The arrow on the second rupture indicates the westward super-shear rupture[4–6,8,9,77]. The shaded relief background map in (**e**) was derived from the shuttle radar topography mission (SRTM) 3-arc seconds data[76].

areas can be observed, indicating near-vertical fault planes dominated by strike-slip. The largest horizontal offset, about 7.8 m, is found along the Pazarcık fault section, close to the junction between the Narli fault and EAF. Two more fault-offset maxima are seen along the Amanos fault and Erkenek fault sections, suggesting respectively about 5.0 m and 6.0 m of slip (Supplementary Fig. 9). Although the second mainshock was characterized by stronger vertical deformation compared to the first, the overall deformation pattern is still dominated by strike-slip movement. The maximum offset is about 8.0 m, near the epicenter, with another offset peak of about 3.0 m along the Maraş fault. From our data, the general trend of fault-parallel displacements along profiles perpendicular to the rupture is first a progressive increase from the far field towards the rupture, consistent with the deformation pattern expected from a dislocation in an elastic medium (Fig. 2d). At some point, however, the fault-parallel displacements depart from the

elastic prediction and decrease when getting closer to the rupture (Fig. 2d). The difference between the predicted slip, according to an elastic deformation model, and the actual fault offset observed at the fault, corresponds to the absent surface displacement (ASD).

## Absent surface displacement (ASD)
We estimated the ASD along the two ruptures by analyzing the fault-parallel displacements of fault-perpendicular profiles (Fig. 2, Supplementary Movie 1, and "Methods"). The larger ASD values, as large as 2 m, are mostly found where the fault ruptures have clear bends or stepovers (Fig. 2e). The ASD ratio, i.e., the ASD normalized by the total deformation (Fig. 3a), is particularly large at the ends of ruptures, as expected, because there the rupture did not reach the surface, although the fault has slipped at depth. Interestingly, the ASD ratio is asymmetric in these areas, with larger values on the compressional side. The reason

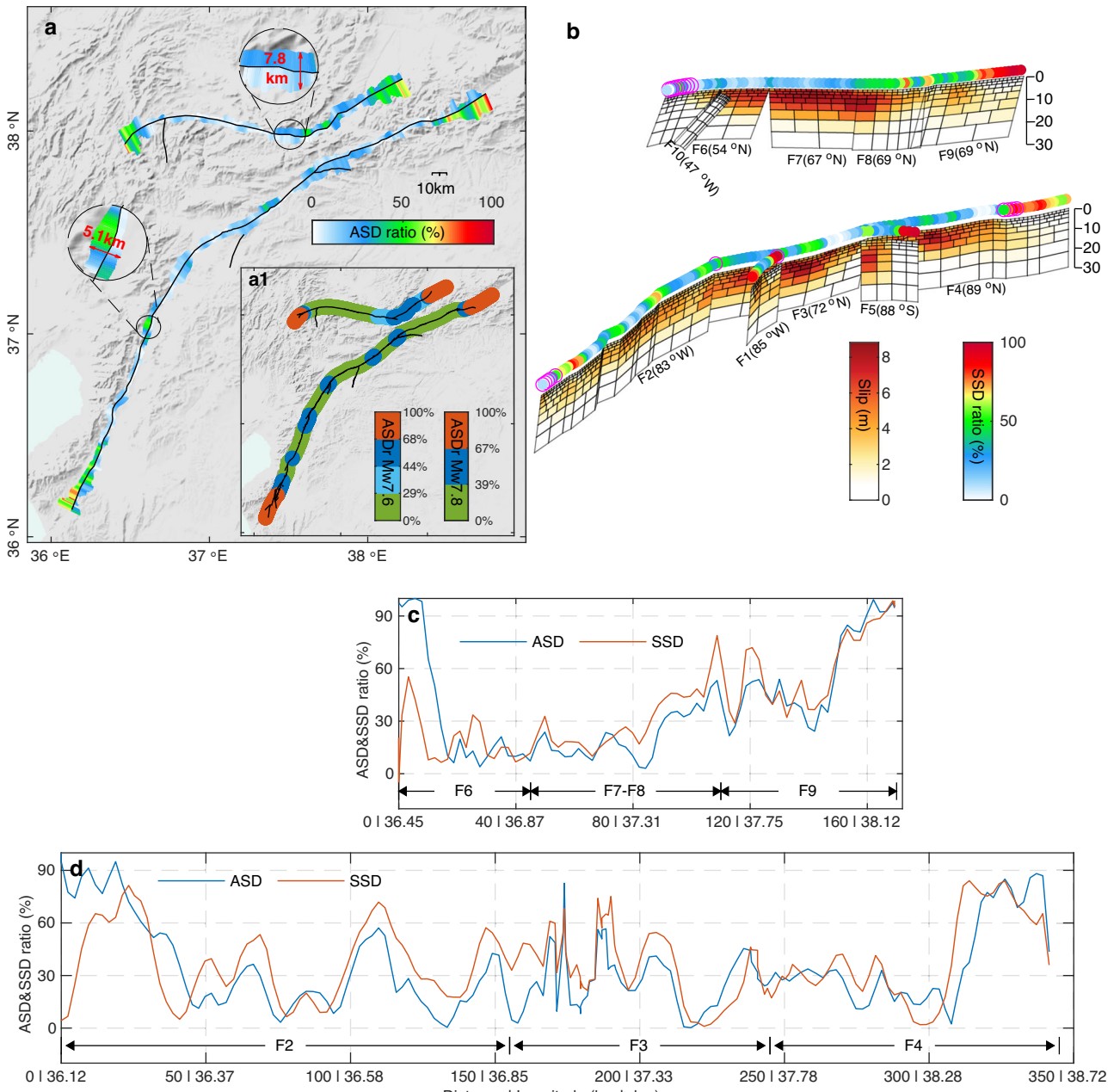

**Fig. 3 | Correlation between ASD and shallow slip deficit (SSD). a** The ASD ratio (ASDr = ASD/total deformation) with respect to the total deformation and ASD extent on each side of the two faults. Two zoom-in areas (black circles) indicate the ASD width, corresponding to the damage width depicted in Fig. 2d, which includes both sides of the fault. The inset shows the classification of the ASDr based on ASDr histogram peaks (Supplementary Fig. 10). Black lines mark surface ruptures[78]. The classes of ASDr values along the ruptures are plotted as thick lines to better visualize segmentations. **b** The fault-slip model with colored circles on the top showing the SSD ratio, which was calculated as "1 − (topmost slip)/(maximum slip at depth)". Magenta circles mark locations with |SSD ratio−ASD ratio| > 40%. **c, d** ASD and SSD ratios along the main ruptures of the Mw 7.6 and Mw 7.8 events, respectively. The shaded relief background map in (**a**) was derived from the shuttle radar topography mission (SRTM) 3-arc seconds data[76].

could be that as rock deforms more easily under extension, it localizes the deformation, whereas in compression more diffuse bulk deformation results in larger ASD ratios. In general, the ASD ratio is smaller than 67%, and the mean ASD ratios are respectively 33% and 36% for the Mw 7.8 and Mw 7.6 events. Those averages, however, are pulled up by significantly larger values found in zones of geometrical complexity and at both ends of each rupture. Still, this suggests that when geologic slip rates are determined using geomorphic offset measurements along the fault trace, the resulting slip rate could be underestimated by as much as a third, depending on the site measurement, and lead to underestimation of the earthquake hazard.

The average widths of the ASD zones are respectively 5.3 km and 6.4 km for the two events. However, the ASD width and amplitude are far from uniform along the two ruptures (Supplementary Fig. 10). The ASD ratios and widths are smaller along geometrically smooth sections of the ruptures and larger in areas of geometrical complexity (i.e., at bends, step-overs, and junctions of fault branches) (Fig. 3a).

### Kinematic slip model
We constructed a five-section fault model of the two events with a fault geometry consistent with mapped ruptures from our SAR-based pixel offset tracking observations. The values of the dip for each fault

section were determined by a grid search (see "Methods"). We then estimated the fault slip (Fig. 3b), assuming an elastic dislocation fault model, by inverting quadtree-downsampled 3D displacements[34] using finer fault-slip patches near the surface to reflect the fault-slip resolution and to better reveal the shallow slip distribution[4]. The overall slip variations are consistent with previous results[4,10], with the maximum slip at a depth of ~5 km. The slip model has three slip asperities with a slip of 5.0 m, 8.5 m, and 8.0 m (from SW to NE) for the first event, and two asperities of 8.5 m and 4.0 m for the second event, which correspond to observed maxima of the surface deformation (Supplementary Fig. 9).

The estimated slip of the topmost patches is generally lower than that at depth, resulting in shallow slip deficit (SSD) along most of the ruptures. Different from the existing studies that calculate the average SSD ratio based on only one curve between the average slip along the fault strike and depth, we calculated the SSD ratio along the faults (Supplementary Fig. 11) to illuminate spatial variations of the SSD. Like the distribution of ASD ratios, the SSD ratios show the largest magnitude at the rupture ends, followed by fault areas with geometrical complexities, and the smallest along smooth fault sections. A clear correlation[35] is thus observed between the ASD and SSD ratios (Fig. 3c, d) and also somewhat between the ASD width and SSD depth (Supplementary Figs. 12 and 13). A correlation between the ASD and SSD ratios is expected since the real reduced fault-parallel displacements near the fault (i.e., ASD) will lead to less modeled on-fault slip near the surface than below (i.e., SSD) when slip is determined in the framework of an elastic inversion. Although it is possible to overestimate slip values by ~20% at depths of ~5 km when using a simple homogenous elastic inversion model[36], this overestimation has a negligible effect on the correlation analysis between the ASD and SSD ratios (see Supplementary Discussion 1 and Supplementary Fig. 14).

## Discussion

To distinguish whether the observed ASD originates from a true reduction in shallow fault slip, as suggested by the elastic SSD modeling, or from off-fault damage, we first assume the former explanation is true. In this case, the total slip accommodated by the fault, over the entire earthquake cycle, should be the same over the seismogenic part of the fault, i.e., from the surface down to the bottom of the fault locking depth[14]. This means the observed coseismic ASD would have to be eventually compensated by shallow interseismic slip (e.g., aseismic creep and transient slow-slip event) or postseismic afterslip[37,38]. The moment of the estimated SSD needed to compensate for the observed ASD is $5.65 \times 10^{19}$ Nm, corresponding to a Mw 7.1 event (Supplementary Fig. 16). However, SAR-based interseismic deformation mapping[39], based on data from 2014 to 2019, indicates no surface creep on the fault sections activated in the Kahramanmaraş earthquakes (Fig. 4a), which rules out shallow fault slip in the years before the earthquakes and suggests that creep is not common along the ruptured sections of the EAF in general. Even if transient slow-slip events did occur before the InSAR observation period, their magnitude would typically be only a few millimeters, with recurrence intervals of several years[40,41]. It is thus unlikely that transient slow-slip events compensated for the SSD of 2–4 m during historical times without being detected.

We also assessed the amount of shallow afterslip using the first nine months of postseismic Sentinel-1 SAR images. Although we cannot rule out that some part of the postseismic deformation might be masked due to the superposition of the deformation from the two events that partially cancels out the deformation in the region between the two ruptures, it appears that most parts of the main ruptures are free of near-fault postseismic displacements (Fig. 4b and Supplementary Fig. 17). Indeed, shallow afterslip only exists in specific locations (e.g., at the northeastern end of the first rupture[42]) and the amplitude does not exceed few tens of centimeters (Supplementary Fig. 18), far less than is needed to catch up with the modeled meter-scale SSD.

Therefore, while we cannot exclude the possibility of shallow slow-slip events before 2014, the current interseismic and postseismic data do not support that shallow slip deficit is responsible for the observed ASD.

Conversely, there is ample evidence for off-fault damage being responsible for the observed ASD. For example, distributed surface ruptures and fringe discontinuities in L-band ALOS-2 interferograms provide evidence for off-fault damage. The inset panel in Fig. 3a shows a strong correlation between the distributed small ruptures and the relatively high ASD ratio (i.e., the light blue and blue colors). While the longer wavelength of L-band ALOS-2 interferogram is capable of tracking the high deformation gradient near the fault, the off-fault inelastic deformation would involve movements like warping, rigid-block rotation, and localized small-scale fracture movements that will lead to near-fault discontinuous or distorted fringes in ALOS-2 interferograms (Fig. 4d–f). Hence, we suggest that the majority of the observed ASD along both ruptures of the Kahramanmaraş earthquakes is related to actual off-fault secondary fault offsets and inelastic volumetric deformation, rather than to shallow elastic slip deficit. Unwrapped ALOS-2 interferograms would allow for differentiating between secondary fault offsets and inelastic deformation within observed ASD areas. However, unwrapping interferograms near coseismic fault ruptures of large earthquakes is usually challenging.

The ASD ratio represents the proportion of the ASD relative to the total deformation, thus it can be used as an indicator of the level of off-fault damage. Figure 3a shows that the off-fault damage has clear spatial characteristics with stronger damage in fault sections that are geometrically complex than in sections that are relatively straight. The average ASD ratio for the straight sections (green in Fig. 3a1) is 20% and 14% for the first and second events, respectively, and 50% and 43% for the sections with geometric complexity (blue in Fig. 3a1). Geometric complexities of faults often act as fault rupture barriers between different fault segments with the rupture energy partly inelastically consumed as the rupture proceeds to the next segment[43,44]. This is probably the reason for stronger off-fault damage where the fault is geometrically complex. While other factors, such as the type of near-surface materials (sediments vs bedrock)[18], fault dip (Supplementary Fig. 13)[27], and rupture speed[43,45], may influence the level of the off-fault damage, geometrical fault complexities appear to be the controlling factor in the Kahramanmaraş earthquake case.

The main difference between earlier earthquake studies based on optical images (Supplementary Table 1) and our result is that the obtained fault damage width of previous cases is clearly smaller than our result of 5–7 km, even for earthquakes of similar magnitude as the Kahramanmaraş earthquakes (e.g., the 2013 Mw 7.7 Balochistan, Pakistan earthquake[17,27]). This is likely due to the narrower aperture of the displacement field used in fault damage analysis of previous studies (i.e., hundreds of meters to several kilometers, Supplementary Table 1) compared to the 100-km-wide SAR-based displacement field used here. Calculating ASD involves extrapolating observations to determine the total deformation. Therefore, previous cases with smaller apertures of displacement are only sensitive to ASD at spatial scales much smaller than the width of the input dataset. This limitation prevents them from precisely deciphering the total far-field elastic deformation curve across the fault (i.e., the red model curve in Fig. 2d), even though the optical images are able to reveal the entire width of the deformation pattern of earthquakes[46–48]. To explore the decay of ASD with distance from the rupture, we calculated the cumulative ASD with zone width across the Kahramanmaraş ruptures. Our results reveal that about 95% of the ASD occurred within distance of the average ASD width (i.e., 5.3 km and 6.4 km for two events) and the ASD within about 1.5 km range accounts for 50% of the total ASD (Fig. 5). This suggests that the ASD in previous earthquake studies with ample geometrical fault complexities, such as of the 2019 Ridgecrest earthquakes[19,49], may be underestimated, as only 1–2 km ASD width

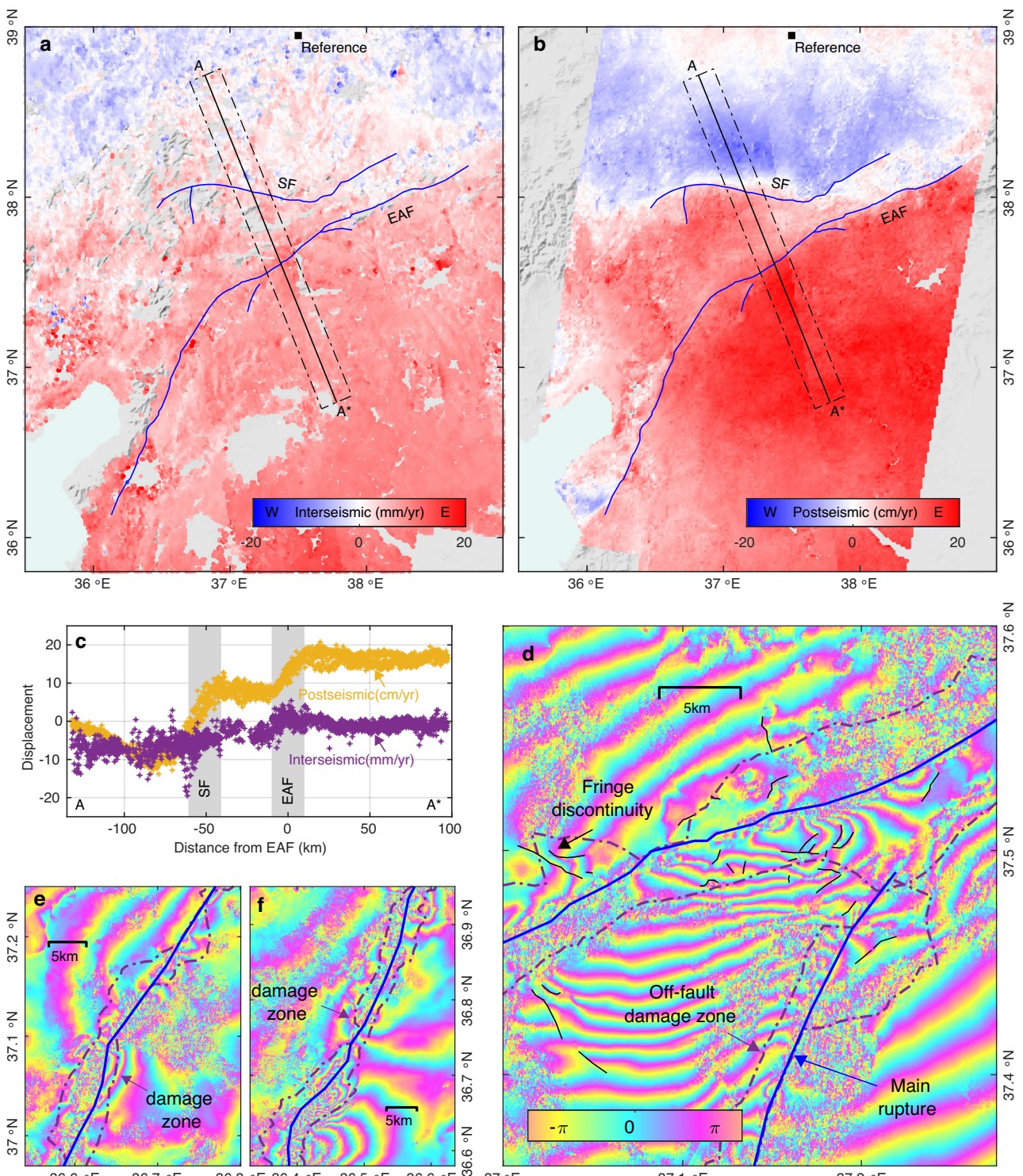

**Fig. 4 | Evidence for observed ASD originating from off-fault damage. a** The east−west interseismic displacement rate in the study region from InSAR time-series of 2014−2019 Sentinel-1 images, with a black square marking the reference point. Blue lines are the two fault ruptures on the Sürgü fault (SF) and the east Anatolian fault (EAF). Displacements along profile A−A* are shown in (**c**). **b** The east−west postseismic displacement rate obtained from 9 months of Sentinel-1 SAR images after the earthquakes. **c** Inter- and post-seismic displacement rates along profile A−A*, which are insufficient to compensate for the observed meter-level coseismic ASD. **d** ALOS-2 SAR interferogram (track 77) of the Narli fault and EAF junction showing distorted fringes and fringe discontinuities within and around the off-fault damage zone. **e**, **f** Two more examples of distorted fringes within the off-fault damage zone. These distorted and discontinuous fringes illustrate that off-fault damage (i.e., inelastic deformation and secondary fault offsets) occurred in these areas. The coverage of (**d**−**f**) is indicated in Fig. 2e, and a clear version of (**d**−**f**) without labels is shown in Supplementary Fig. 15. The shaded relief background map in (**a**−**b**) is derived from the shuttle radar topography mission (SRTM) 3-arc seconds data[76].

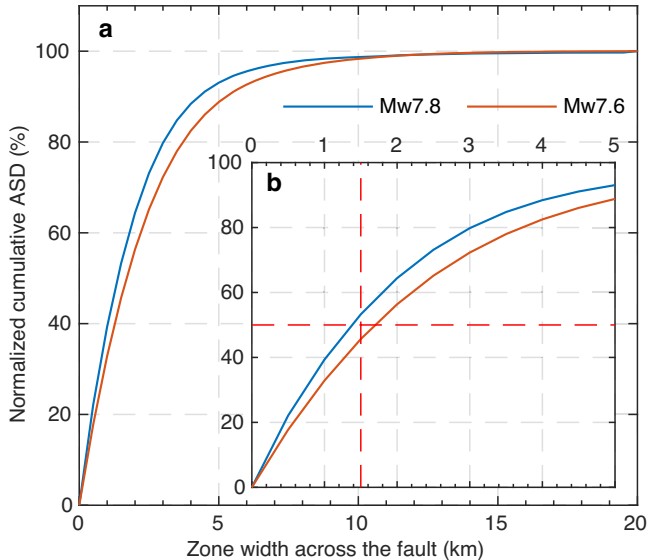

**Fig. 5 | Normalized cumulative ASD with zone width across the fault.** For
**a** 0–20 km zone width and **b** 0–5 km zone width. Note that a 5 km zone width of the
*x*-axis refers to 2.5 km on each side of the fault. Supplementary Fig. 19 illustrates the
process of calculating the normalized cumulative ASD for a specific zone width.

was detected[19]. This underscores the significance of utilizing large-
enough remote sensing images to investigate off-fault damage and
extending observations of off-fault damage beyond 5 km away from
fault ruptures.

Field investigations and seismic experiments also show relatively
narrow zones of off-fault damage[20,50], typically much less than the
~5 km reported here. The main reason for this discrepancy is that off-
fault damage may manifest as inelastic rock deformation (e.g., warping
and rigid-block rotation), which might not create obvious surface
features (e.g., ruptures) or lead to significant rock failure within the
bulk medium. Consequently, accurately identifying the extent of the
off-fault damage zone through field investigations or seismic wave
velocity estimation is challenging. Additionally, it has been shown that
coseismic damage zones undergo a healing process afterward, as
indicated by the increased seismic velocities around the fault during
the postseismic phase[51–53]. Hence, following this healing process, the
medium within the damage zone is similar to the bulk medium (i.e.,
after the geometric closure of opened cracks[53]). Here, we offer a geo-
detic perspective on coseismic off-fault damage, which should be
integrated into geodetic and seismic datasets in both coseismic and
postseismic analyses for future earthquakes.

In this study, we have reported on extensive off-fault damage
around the main ruptures of the 2023 Kahramanmaraş earthquakes
based on accurate and large-scale SAR-based coseismic 3D displace-
ments. By jointly analyzing the interseismic, postseismic, and coseis-
mic deformation, we attribute the observed ASD to off-fault damage
rather than to the reduction of shallow fault slip within the elastic
medium (i.e., SSD). The results indicate that shallow off-fault damage
consumes on average about 35% of the on-fault coseismic slip at depth.
This fraction is lower on fault sections that are relatively straight and
free of geometrical complexities. Still, due to this fault damage, near-
fault slip-rate estimates could be underestimated by 10–20% on con-
tinental strike-slip faults and possibly as much as a third. This should
also be accounted for in paleoseismological works where it is not
always possible to find an ideal site near a structurally simple segment
with limited off-fault damage. In addition, we find the average width of
the off-fault damage zone to be ~5 km, significantly wider compared
with previous studies reporting damage widths of only a few hundred
meters. The spatial heterogeneity of the observed off-fault damage,

coupled with its distribution over wider areas, calls for a reconsi-
deration of slip-rate studies and seismic hazard assessments along
major faults.

## Methods
### SAR data processing
**Coseismic SAR data processing.** We used five pairs of ALOS-2 and
three pairs of Sentinel-1 SAR images (Supplementary Table 2) to obtain
coseismic displacement observations from interferometric SAR
(InSAR)[54], multiple aperture interferometry (MAI)[55], pixel-offset track-
ing (POT)[56], and range split-spectrum interferometry (RSSI)[57]. These
data were processed using the GAMMA software and the shuttle radar
topography mission (SRTM) digital elevation model (DEM) was used to
assist the image coregistration and to remove the topography com-
ponent in the interferograms. The Sentinel-1 and ALOS-2 SAR images
were multi-looked by 10 × 3 and 3 × 15 (range × azimuth), respectively,
yielding a final pixel size of ~50 m × 50 m. The InSAR data were filtered
with an adaptive filter[58] using a filter window size and exponent of 32
and 0.4, respectively. This was followed by minimum cost flow inter-
ferogram unwrapping after manually and iteratively masking out low
coherence areas. For MAI, the full-aperture SAR images were separated
along the azimuth direction into backward- and forward-looking SAR
images, which were then used to generate backward- and forward-
looking interferograms based on the InSAR procedure. The MAI
interferograms were then produced by differencing these two InSAR
interferograms. For the TOPS-mode Sentinel-1 SAR data, we conducted
the MAI procedure for each burst and then stitched all burst results
together into a complete MAI interferogram. For the POT processing,
we applied coregistration windows of 32 × 192 and 128 × 32 (range ×
azimuth) pixels to obtain pixel offsets from the ALOS-2 and Sentinel-1
SAR images. These windows correspond to about 500 m × 500 m
(ground range × azimuth) on the ground, therefore the observed ASD
with up to 5 km width is unlikely to originate from the window-based
pixel coregistration process (Supplementary Fig. 20). We did not use
larger coregistration windows for the POT processing since larger
windows could smoothen out sharp displacement offsets across faults
and bias our off-fault damage analysis (Supplementary Figs. 21 and 22).
The RSSI technique is sensitive to the same deformation component as
InSAR, but capable of measuring larger deformation gradients, e.g.,
near earthquake fault ruptures where InSAR fringes are too dense. The
RSSI procedure is similar to that of MAI, except the range spectrum (as
opposed to the azimuth spectrum) of the original SAR images is split in
two, producing two sets of SAR images, from which upper- and lower-
band interferograms are formed, and then differenced to yield the
RSSI observation. For each SAR-image pair, we can thus retrieve
coseismic displacements along the slant-range direction using InSAR,
RSSI, and POT and along the azimuth direction using MAI and POT.
Therefore, a total of 40 displacement observations can be generated
with eight pairs of SAR images (Supplementary Fig. 5).

**Postseismic SAR data processing.** To study the postseismic defor-
mation, we used 9 months of Sentinel-1 SAR data from after the
earthquakes. The data were acquired from one descending and two
ascending tracks (same as used for the coseismic deformation, Sup-
plementary Fig. 4). To minimize decorrelation noise, each SAR image
was combined with the two subsequent SAR acquisitions to generate
short temporal-baseline interferograms (Supplementary Fig. 23). After
adaptive filtering and phase unwrapping, we inverted for displacement
time series using these three sets of short temporal-baseline
interferograms[59]. Finally, we combined the three InSAR displacement
rate maps (Supplementary Fig. 17) to derive the near-east (Fig. 4b) and
vertical (Supplementary Fig. 24) displacement rates, omitting possible
contributions from north displacements. Furthermore, since the InSAR
line-of-sight observations are insensitive to the north–south displace-
ments and thus reveal less about the postseismic deformation pattern

near the southwestern segment of the Mw 7.8 ruptures, we also obtained azimuth postseismic displacement rates in burst-overlap areas (Supplementary Fig. 17) of the Sentinel-1 images using burst-overlap InSAR (BOI)[60,61].

## Coseismic 3D surface displacements

The coseismic 3D surface displacements were calculated with a method based on the strain model and variance component estimation (SM-VCE)[31,32]. In the SM-VCE method, a strain model[62], representing the geophysical relationship between the 3D deformation of adjacent points, is employed to establish the Green function between SAR-based displacement observations and 3D displacements. In this case, SAR observations around a target point can be used to estimate the 3D displacement components of this point, which is more robust than the standard weighted least squares method in which only the SAR observations of the target point are used in the calculation. Furthermore, the SM-VCE method weighs the different SAR observations in a posteriori and iterative way based on the classic variance component estimation algorithm[63]. Benefiting from the incorporation of the strain model and variance component estimation algorithm, the SM-VCE method has been proven to be superior to the standard weighted least square method[64,65] by obtaining more accurate 3D displacements in several case studies, such as for the 2022 Menyuan (China) and 2019 Ridgecrest (CA)[66] earthquakes. Even if only the noisier POT observations are available in the near-fault areas, the 3D displacements can still be well resolved, e.g., for the 2021 Maduo (China)[67] and 2016 Kaikoura (New Zealand)[68] earthquakes. Here we combined twenty-three independent SAR displacement observations (Supplementary Fig. 5) to calculate the 3D displacements with the SM-VCE method. In the near-fault areas, up to 12 independent POT-based observations are available, providing sufficient data for reliable 3D displacement derivation. To establish the Green function with the strain model, observations are considered within a window of 15 × 15 pixels (750 m × 750 m, about 50 m for each pixel), which is much smaller than the damage-zone width of 5 km obtained here. However, when the target point is close to a fault rupture, it is unreasonable to include observations from both sides of the rupture to establish the Green function. Thus, in these cases, we eliminated the observations from the other side of rupture based on a strain model-based adaptive neighborhood determining algorithm[68].

Note that although our SM-VCE-obtained coseismic 3D surface displacements look smooth, this smoothness comes more from the employment of the strain model in the SM-VCE method, rather than adding any mathematical form of regularization during the calculation. The rationale of the strain model lies in that the deformation gradient within a window (i.e., 750 m × 750 m in this paper) is a constant and the deformation value changes smoothly within this window (Supplementary Fig. 25).

## Absent surface displacement (ASD)

In the absence of inelastic deformation, the surface displacement $L(x)$ across a fault over one full earthquake cycle (Fig. 2a) is expected to be like block motion, which can be expressed as:

$$L(x) = \begin{cases} -0.5s_0 & x<0 \\ 0.5s_0 & x>0 \end{cases} \qquad (1)$$

where $x$ is the distance from the fault and $s_0$ is the relative offset between the two sides of the fault. The interseismic part of the elastic displacement $I(x)$ can be modeled by the *arctan* function (Fig. 2b):

$$I(x) = \frac{s_0}{\pi} \cdot \arctan\left(\frac{x}{D}\right) \qquad (2)$$

where $D$ is the locking depth. In this case, the expected elastic coseismic displacement $C(x)$, ignoring postseismic deformation, can

be obtained by differencing $L(x)$ and $I(x)$ (Fig. 2c):

$$C(x) = \begin{cases} -0.5s_0 - \frac{s_0}{\pi} \cdot \arctan\left(\frac{x}{D}\right) & x<0 \\ 0.5s_0 - \frac{s_0}{\pi} \cdot \arctan\left(\frac{x}{D}\right) & x>0 \end{cases} \qquad (3)$$

However, due to the off-fault damage that consumes part of the coseismic deformation close to the fault (i.e., the ASD), the real observed coseismic displacement decreases in magnitude in the off-fault damaged zone (Fig. 2d). Therefore, we define the ASD as the residual between the coseismic displacement observations with respect to the predicted elastic coseismic displacement, which is obtained by fitting Eq. (3) to displacement observations beyond 5 km away from the fault. We selected this 5 km threshold as the width of the off-fault damage zone is mostly smaller than 5 km, as shown in Supplementary Fig. 10. Note that it is not necessary to determine the relative offset $s_0$ for Eq. (1) and the locking depth $D$ for Eq. (2) in advance; these two parameters are taken as unknowns in Eq. (3) and are estimated when fitting Eq. (3) to the coseismic displacements. Although the implicit assumption of Eq. (3) is uniform coseismic slip within the seismogenic depth, our Supplementary Movie 1 demonstrates the capability of Eq. (3) in fitting the coseismic displacements, indicating a negligible effect of this assumption on the analysis conducted here. Figure 2e shows the estimated ASD, i.e., the lack of expected elastic coseismic displacements (yellow dots in Fig. 2d). As can be seen, the ASD is at a maximum near the fault and then decays to zero with distance. Based on this characteristic shape, we used the following logarithm function to fit the ASD observations:

$$S(x) = a \cdot \log_{10}\left(\frac{x + c_0}{x_0 + c_0}\right) \qquad (4)$$

where $c_0$ is a constant to prevent bracket values close to zero, $a$ is an amplitude adjustment parameter, and $x_0$ is the extent of the off-fault damage (i.e., the ASD width, red squares in Fig. 2d). Based on Eqs. (3) and (4), we thus derived model curves, ASD widths in Fig. 2d, total deformation, on-fault offsets, the ASD, and the ASD ratio from the multiple fault-perpendicular profiles (of fault-parallel displacements) along two earthquake ruptures.

The above calculations are based on fault-parallel displacements of each fault-perpendicular profile. Each fault-perpendicular profile is oriented perpendicular to the local fault strike and extends 50 km away from the fault on each side. We select points along each profile every 50 meters (given the spatial resolution of the 3D displacements is ~50 m) and calculate the displacement values at these points by averaging coseismic east and north displacements of the nearest four neighbors, excluding neighbors on the opposite side of the fault. Finally, the fault-parallel displacement is obtained by projecting the averaged horizontal displacements onto the profile.

By fitting the fault-parallel displacements along fault-perpendicular profiles with Eqs. (3) and (4), we can obtain the magnitude and extent of the off-fault damage. However, it is difficult to accurately determine the internal structure details within the damage zone based only on the SAR-based observations since the observed ASD within these regions is just one geodetic manifestation that could be captured by SAR images. Future investigations combined with other observations (e.g., ground penetrating radar) are required to shed light on the near-field structure details.

Authors of earlier fault-damage studies calculated ASD in a different manner[19,47,49,69,70]. It is well known that coseismic deformation observations[15,71,72], outside near-fault regions, clearly show elastic-like arctangent shape of the coseismic displacements for strike-slip events (red dashed lines in Supplementary Fig. 26c), increasing from the far field towards the near-fault areas. Previous studies assumed that the surface displacement decreases within the damage zone, consistent

with our assumptions. However, they considered the displacement values outside the damage zone to be constant or vary linearly with the distance (Supplementary Fig. 26a), whereas we use the elastic arctangent function to fit the coseismic displacement outside the damage zone. By combining the expected elastic deformation with the observed decreased displacement in the near-fault damage zone (blue lines in Supplementary Fig. 26c), our approach straightforwardly derives the ASD, avoiding the assumption of constant displacement outside the damage zone (blue lines in Supplementary Fig. 26a), which substantially underestimates both the total slip and the ASD. Furthermore, while previous studies analyzed off-fault damage using near-fault observations limited to hundreds of meters to a few kilometers from the fault (Supplementary Table 1), our study uses 100-km-long displacement profiles, allowing us to capture the large-scale signal of off-fault damage.

### Finite-fault inversion

The fault trace was determined from the SAR POT results and the model fault was constructed from the surface to a depth of 30 km. Both the Mw 7.8 and Mw 7.6 events were modeled using five fault segments (Fig. 3b), each discretized into multiple rectangular patches. The patch width (i.e., height) increases with depth by a factor of 1.35 for each row from 1.2 km at the top and the patch length is roughly twice of patch width. This is because SAR displacements resolve better shallow fault slip than deep slip[4,10,33,73]. We assume a planar fault geometry for each fault segment, where the topmost trace of the fault plane is fixed to the main fault trace, as determined by sharp offsets in the near-field POT results. The best dip angle of each fault segment was determined by a grid-search strategy (Supplementary Fig. 27). Although there may be small secondary fault ruptures within the ~5 km wide damage zone, apart from the main rupture, these are difficult to detect and incorporate into the slip model. By using planar fault segments aligned with the surface fault trace, the modeled surface displacements match well with our observations, indicating that the main fault plane within the seismogenic depth used in the fault slip inversion is well constrained. Since patches are unevenly distributed on the fault plane, we imposed inverse distance-weighted smoothing constraints between adjacent patches, which means that the inversion procedure favors slip on a target patch to be similar to the inverse-distance weighted average slip of the surrounding patches (Supplementary Fig. 28). A smoothing factor of 10 was selected based on a trade-off curve between the normalized slip roughness and observation misfit (Supplementary Fig. 28). Using this setup, quadtree-downsampled 3D displacements (Supplementary Fig. 29) were used to invert for the fault slip distribution in a uniform elastic half-space[12], in which the surface fault trace is taken as additional input to exclude the points on the other side of the fault for quadtree blocks close to the fault trace. Based on the 3D displacement observations, dip-slip was set to zero for the first event. For the second earthquake, segment F9 was set to have a pure thrust mechanism, segment F10 to have a right-lateral strike-slip, while other fault segments were to have a left-lateral strike-slip and normal faulting. The resulting strike slip and dip slip solutions are shown in Fig. 3b and Supplementary Fig. 30, respectively.

## Data availability

The Sentinel-1 data were downloaded from the Alaska Satellite Facility (https://vertex.daac.asf.alaska.edu/) and the ALOS-2 data are from the Japan Aerospace Exploration Agency (https://www.eorc.jaxa.jp/ALOS/en/dataset/alos_open_and_free_e.htm). The post-processed data presented in this study have been deposited in the Figshare database at https://doi.org/10.6084/m9.figshare.25801234. The data supporting the findings of this study are provided in Supplementary Information and Supplementary Movie 1.

## Code availability

The SM-VCE method used in this study is publicly available at https://zenodo.org/records/6346205. All other calculation codes and examples used in this study are available on request from the corresponding author.

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

## Acknowledgements
We are grateful to Professor Roland Bürgmann and Professor Tim J. Wright for their constructive discussions. This work was supported by King Abdullah University of Science and Technology (KAUST) (grant BAS/1/1353-01-01, SJ).

## Author contributions
Conceptualization: J.L., S.J., and Y.K. Investigation: J.L., S.J., X.L., W.Y., and Y.K. Methodology: J.L., S.J., X.L., and Y.K. Project administration: J.L. and S.J. Supervision: S.J. and Y.K. Writing-original draft: J.L. Writing-review and editing: J.L., S.J., X.L., W.Y., and Y.K.

## Competing interests
The authors declare no competing interests.
