## [Transparent Peer Review file · Nature Communications]

Extensive off-fault damage around the 2023 Kahramanmaraş (Türkiye) earthquake surface ruptures

Corresponding Author: Dr Sigurjon Jonsson

Version 0:

Reviewer comments:

Reviewer #1

(Remarks to the Author)

This manuscript focuses on the 2023 Kahramanmaraş earthquake sequence in Turkey, using InSAR geodesy to constrain the 3D displacement field spanning both the interseismic, coseismic, and early postseismic period. The authors examine the near-fault displacement pattern, and attempt to characterize the style of deformation around the fault surface rupture. They conclude that a significant proportion of deformation (~35%) is accommodated within a diffusely deforming damage zone, extending on average 5 km either side of the rupture, which generally increases in width close to geometric complexities (e.g. fault step-overs, etc). The main conclusion is that these off-fault damage zones may accommodate deformation over wider regions than typically considered in studies employing optical image correlation data, and that on-fault coseismic slip may not be representative of the total surface offset accommodated across the entire fault zone during an earthquake.

The paper is well-written, and contains much useful additional information in the supplementary material, although I did find it a bit complicated at times going back and forth between the main paper, the methods section, and the supplementary figures (which are important to the story in some cases). The topic is of interest, and the conclusions somewhat provocative, with potential interest for the community. However, I didn't always find the interpretations and discussion fully supported by the assumptions made (e.g. regarding ASD), or the data as presented; I have some reservations about some of the interpretation, and questions regarding potential artifacts arising from the 3D displacement inversion. This paper could be strengthened by demonstrating the 3D displacement result is not biased in the near-field (ideally by projecting the 3D displacement back into radar geometry and plotting the residuals in each InSAR dataset), as well as integrating with the optical data, which can give similarly good medium and near-field displacements. For me, the paper currently doesn't conclusively demonstrate that the near-field roll-over of displacement into the fault zone is the result of inelastic deformation within a very wide damage zone. Some of the results are also similar to existing studies looking at the fault slip distribution. But nevertheless, I found the paper interesting and easy to read.

My main comments are below:

1. The authors do not utilise all the available geodetic data, ignoring optical correlation measurements which could significantly improve their 3D displacement dataset.
2. Neither do the authors demonstrate how well their 3D displacement result (which is obtained through an inversion procedure) fits the raw data.
3. The methodology used to obtain the displacement is not discussed in much detail (referring the reader to another more technical paper in IEEE TGRS). However, the two papers cited to justify this 3D inversion technique (Liu, et al., 2018, 2022) both integrate InSAR data which extends close to the fault rupture, and therefore do not rely on the much noisier pixel offset tracking data to constrain the nearfield. However, in this Turkey earthquake sequence, the near-field offsets are largely constrained by pixel offset tracking, since the InSAR data does not extend anywhere near the rupture for Sentinel-1, and is even quite far for ALOS-2.
4. The authors do not give many details about the unwrapping technique they use, which is probably less robust than, say, the branch cut method, which may be better for dealing with discontinuities.
5. Given the above comments, it remains unclear what bias is introduced to the near-field displacement results using their inversion procedure, which is heavily weighted towards the extremely noisy pixel offset tracking results. The inversion procedure relies on some form of regularization, I don't know to what extent this impacts the near-field displacements? This,

for me, may be critical, because the scientific novelty of this paper is focused on the near-field displacement data.

6. The correlation window also uses relatively small window sizes for the range direction (ALOS-2) and azimuth direction (Sentinel-1), which may in fact be too small to obtain robust displacements in these components; when the window size is too small, there is not enough information to obtain a highly accurate result, thus imparting a sub-pixel bias which tends toward zero; this effect is commonly seen with optical correlation data when using windows too small given the level of noise in the image windows (unless mitigated by spatial regularization, but this can also smooth out discontinuities). Given that amplitude images are relatively noisy at high frequencies (due to the speckle), it's unclear if 32 pixels is appropriate for resolving an unbiased displacement in the range direction (ALOS-2) and azimuth direction (Sentinel-1).

7. The fault-parallel displacement profiles in Fig. 2 appear highly smooth in the near-field region. I compared these with results from correlation of Sentinel-2 data, and they look overly smooth in the near-field region. Also, the residuals (once the model results have been removed) show spatial variability with length-scales of ~5 km, which suggests maybe some level of smoothing is present in this displacement data (presumably resulting from the 3D inversion method?).

8. Since these near-field displacement measurements are critical for the main story of this paper (that damage zone widths extend further than previously thought), I would encourage the authors to perhaps (1) supplement their results with those from optical correlation data, which provides an independent and more detailed view of the near-field displacement), and (2) to see how well their 3D displacement solution accounts for the information in their raw displacement data; for example, projecting the 3D displacements back into radar geometry and plotting the residuals would be important to convince the reader that the near-field displacements are well resolved. Such an endeavor should also show appropriately zoomed in views of the near-field region, as it would be challenging to see at the scale shown in Fig. S3.

9. I found the section on calculating ASD slightly confusing. I follow the concept that a long-term block-like offset minus the interseismic can give you the co-seismic (plus postseismic, and any potential interseismic creep bursts, which are somewhat ignored here). However, the interseismic observations do not cover the entire interseismic period, therefore how do you know what the total displacement should be, i.e. the magnitude of the long-term offset (the step in Fig 2a)? Maybe you compute the long-term step over the same time-period as the interseismic observations? But this would ignore any temporal variations in the interseismic strain accumulation, which could be possible. Also, this approach assumes the coseismic has a uniform down-dip displacement, which you demonstrate later with your subsequent slip inversion (Fig. 3b) not to be the case.

10. The profile chosen in Fig. 2d is maybe not a good/representative case, because it is located where the Narli fault intersects the main EAF. Therefore, you have chosen a place where the fault structure is more complicated, and thus not well represented by the simple block model shown in Fig. 2a. Indeed, the locations where the fault zone width appears relatively high correlate with locations of structural complexity, and thus perhaps simply a failure of your simple block model to accurately reflect the complexity in the near-field structure.

11. A more minor point, but the variation of the fault strike over the width of the swath profile used to stack the fault-parallel displacement profiles can also artificially widen the apparent fault zone width (see Milliner, et al. 2021) (although I acknowledge probably not to the extent of 5 km).

12. The similarity in the ASD and SSD plots in Fig 3d looks to me like the ASD is simply showing the same thing as SSD. In the case of SSD, we're looking at the difference between peak slip (around 5 km depth) and the surface. Since you assume a simple discontinuity with no constraints shallower than 5 km, it seems unsurprising that you get the same result.

13. Regarding the apparent SSD resolved in your slip model is that maybe this is simply a feature of the data, and that the reduced slip in the shallow crust is well-resolved, and there is no distributed off-fault damage at all? In other words, why does this slip deficit have to be distributed off-fault damage, and not simply tapered slip towards the surface?

14. In the study of Marchandon, et al. (2021), the authors demonstrate that inverting with an overly simplistic model can artificially enhance slip at depths of ~5 km. Therefore, even if the slip at the surface is well-resolved (e.g. by pixel tracking or optical correlation data), one may infer an apparent SSD from the slip model because the slip at depth is enhanced. Could that be the case here, since the slip inversion is made with a homogenous elastic half space with a somewhat simple fault model (given the complexity we see along the fault rupture), etc? In this case, SSD is actually the result of overestimating the slip at depth, rather than underestimating the slip at the surface?

15. How do you decimate your displacement data for use in the inversion? Or rather, how do you avoid smoothing over the fault with your minimum quadtree block size, while retaining enough near-field data to adequately constrain the shallow-most slip?

16. Field measurements of fault damage zones are generally rather narrow (e.g. Mitchell, et al. 2009), much less than 5 km as reported here. Similar observations for dense seismic experiments across fault zones show similar things (Li, et al., 2007). How do you reconcile these observations with your interpretation of such a large damage zone? Also, how do you interpret these damage zones in the context of healing, as seen in Vp/Vs studies (Brenquier, et al., 2008)?

17. In terms of the conclusions, a field geologist would typically be aware of the problems associated with estimating a representative fault slip rate using Quaternary dating of offset geomorphic markers for a structurally complex segment of a fault (vs a structurally simple one). So it's not entirely simple to present this issue in such black and white terms. Ideally, one would estimate a fault slip rate on a simple segment, subject to less distributed off-fault deformation (though it really depends where the offset geomorphic markers can be found). It's maybe more true that trenching studies might be more biased towards structurally complex sections, where vertical movements influence drainage networks, thus facilitating more continuous sedimentation.

18. Line 241 claims that optical is unable to resolve the medium or far field displacement, but this is not the case. Jolivet, et al. (2013), and various other studies since, have shown very well that optical data can be used to capture the attenuation of displacement far from the fault rupture. This might be true with very high resolution optical data, but even Pleiades has a footprint that spans 15-20 km (depending on the incidence), and for an east-west striking fault, these images may capture the attenuation of slip over even greater distances.

19. Line 255. I think these widths of 5 km more likely tell us the limitations of the fault slip inversion at capturing the precise details of the fault rupture within the upper 5 km.

20. Fig. S4. What are the 2 arrow scales in (a)? Where is EKZ1 in (b)? This has a larger offset, and is closer to the fault,

therefore more interesting than the very far-field measurements.

21. Fig. S11. This figure highlights how noisy the pixel offset tracking results are close to the rupture. It seems surprising to see how noisy these data are here, and how smooth the final inversion results are in Fig. 2 in the main paper, given that the pixel offset tracking data provide the main constraints on the near-field displacements.

22. Fig. S15. This idea has already been discussed by, for example, Scott, et al. (2018) and Miliner, et al. (2016). Also, case (a) is typically what is seen over very short length scales, while case (b) is seen even with optical data over length scales of several km.

23. Table S1 is a bit limited, and not very comprehensive.

Reviewer #2

(Remarks to the Author)

Key results

The key message of this research is that the measured shallow coseismic offsets at the fault significantly underestimate the true near-surface deformation. The authors show that for the case of the 2023 Kahramanmaraş earthquakes the observed Absent Surface Deformation (ASD) was about 35% of the on-fault coseismic slip at depth. This result goes a long way towards addressing the shallow fault deficit, which has been a consistent observation for many earthquake ruptures around the world.

Validity

The presented results and conclusions are robust and follow a valid, clearly detailed methodology. The data and method as presented support the conclusions. However, I do have some relatively minor but important methodological points that I think need to be addressed. I detail these in a later section.

Significance

The result presented here is the first case where absent shallow deformation (ASD) has been estimated for a major earthquake. The authors show that ASD for the 2023 Kahramanmaraş earthquakes is on average 35% but could be up to 90% of the maximum modelled slip at depth. This is a significant observation that has important implications for the underestimation of long-term seismic hazard assessment based off earthquake offset measurements alone.

Data and methodology

The primary data used in this paper is openly available radar satellite data from the Sentinel-1 and ALOS-2 satellites. The authors present a robust processing methodology that combines routine time series analysis of long-term deformation with detailed and careful analysis of pseudo-3D deformation using a combination of pixel offsets, multiple aperture interferometry (MAI) and range spectrum split interferometry (RSSI).

Analytical approach

The principle analytical approach used in this work has been to use simple analytical models to estimate the expected coseismic slip at the fault. The authors fit the model to the observed displacement in the far field and determine the difference between the observed and modelled slip near the fault to estimate the absent surface deformation (ASD).

Suggested improvements

I have three main points that I think could do with some additional work/clarity.

1. ASD ratios approaching 60% is very large! In some places this is equivalent to more than 5m of 'missing shallow slip'. If all of this is attributed to off-fault deformation within ~5km either side of the fault, I would expect to see much more evidence of distributed slip. I realise the authors show examples of such slip in Figs 4e-d, but I would ask if the authors have attempted to sum up this distributed slip to see how much of the ASD it accounts for?

2. I think the authors haven't given sufficient attention to the role of shallow afterslip in accommodating some of the missing coseismic ASD. In other well documented earthquakes in Turkiye (e.g. Izmit 1999) we see that shallow afterslip rates can be very significant. For example, Hussain et al., (2016) showed that shallow afterslip probably contributed over 1m of additional slip in the shallow portions of the fault. While this might not detract from the conclusions of this paper, it might reduce the overall ASD fraction. I realise that the authors touch on this in Fig 4a (and associated main text) but I don't think this is sufficient. Perhaps one way to investigate the shallow afterslip in detail is by differencing the displacement time series of pixels close to the fault/near the fault within the ASD width.

3. A fundamental and well-known challenge with the interseismic and postseismic observations on at least the southern segment of the Mw 7.8 rupture is that this fault is not optimally oriented for InSAR deformation measurements. Can the authors be confident that they have captured the full long term and early postseismic deformation here?

Additionally, some minor points are listed below:

1. Lines: 122-123: Figure labelling incorrect. I think you meant Fig. 2d. In general, some of the figures are labelled incorrectly and/or incorrectly referenced in the main text. Please check these.

2. Line 268: Slip rate estimates of... [what?]

3. The discussion presented in between Lines 243 and 260 feels incomplete. There have been plenty of detailed studies of coseismic earthquake offsets on larger spatial scales with both radar and optical datasets (e.g. Napa Valley, Ridgecrest, Kaikoura, Palu). Perhaps a better comparison could be made with these studies in addition to the subset of studies that have used optical images of near-fault deformation.

Clarity and context

The authors provide a succinct summary of the problem and explain the context of their findings clearly and concisely. It would have been good to read a little more about the broader implications of their work alongside a discussion of potential limitations (for example due to the viewing geometry issues suggested above).

References

The references are sufficient and appropriate. Although the authors may choose to include additional citations to cover the suggested additions above.

Reviewer #3

(Remarks to the Author)

Thanks very much to the authors for a compelling and well-written manuscript. This study takes advantage of both Sentinel-1 and ALOS-2 SAR data to calculate the coseismic displacements using dInSAR, MAI, POT, and RSSI methods, and then estimate the 3D coseismic displacements using the SM-VCE approach, for the 2023 Kahramanmaraş M7.8 and M7.6 earthquakes. From the 3D displacements, the authors estimate a metric they call the “absent surface displacement” (ASD), which they estimate all along both fault ruptures. The data indicate that ASD can vary along strike and appears to correlate strongly with fault complexity (e.g., complex bends or stepovers result in higher ASD while simple fault stretches have lower ASD), and that, on average, ~35% of fault slip at depth can get “eaten up” by ASD/inelastic deformation near or at the surface. In addition, the data indicate that looking further afield may be very important to determine an accurate measure of ASD/inelastic fault deformation, and that previous studies likely have not looked far-enough away from the rupture to capture the full measurement of ASD. All of this suggests that previous studies have likely been underestimating fault slip (both by not including the full aperture of ASD/off-fault deformation width, and by focusing only on slip measurements along the fault rupture plane), and therefore potentially underestimating fault hazards.

This was a fascinating paper, and I support publication as I believe it will be an important catalyst for continued discussions about how we even just *measure* how an earthquake has deformed the surface, let alone how that flows into our estimates of earthquake hazards along known fault systems. I have a couple line-by-line comments below, but they are mainly cosmetic suggestions/typo corrections. I think the authors have done their due diligence to provide all of their work and their supporting data for this effort.

Line-by-line comments:

L19: Maybe should be “experienced higher levels of off-fault damage” or “experienced a higher level of...”

L40: I think there may be an article missing, perhaps should be “embedded in “an” homogenous or layered elastic...”

Figure 1: Nice figure, very clear

Figure 2: What are the uncertainties on your profile measurement? (and from that, what are the uncertainties on the estimate of ASD?) In (d), should it be labeled fault-parallel displacement to be specific within the figure? (I see it says it in the caption, but it might be beneficial to label it on the axis too)

Question: How trustworthy are the nearest fault measurements? They seem to be critical in order to make this estimate of ASD, but are there any issues with loss of coherence near the fault trace?

Figure 3: Inside the subset, I am a little confused as to why the binned ratios are plotted as being the same on both side of the fault, whereas in the main 3a figure, the ratios appear pretty different in a couple places (on both sides of the fault – e.g., the NE-most end of the main M7.8 rupture). Are you just plotting the highest percentage from either sides? Maybe I am missing something.

L164: just add an article again, for “with the maximum slip at “a” depth of ~5km”.

Figure 4: In (c), what do the triangles at 50 km from the EAF represent? Are they marking something specific or are they just label pointers? For (e),(f), and (d), can you put the non-annotated versions of the wrapped interferograms, so the reader can look for and see the discontinuities for themselves?

Just a thought: have you considered calculating the phase gradient maps for these? I enjoyed the maps produced by

Xiaohua Xu of the 2019 Ridgecrest, CA sequence (Xu et al., 2020, Science), and they seem to highlight this type of off-fault deformation very clearly.

L201-205: I don't necessarily disagree, but four years of InSAR measurements can't tell you that this fault hasn't crept during its "lifetime", can it? Are there any papers that confirm this observation? How many "creep events" would you need to accommodate this magnitude of deformation?

L212: Add an "a" in "does not exceed "a" few tens of centimeters"

L214: At a postseismic rate of 20 cm/yr, in one year, it would make up 1 meter of displacement – are you suggesting most of this is viscoelastic, and therefore not occurring on the shallow fault section?

L268: "...slip rate estimates of..." of what? Maybe missing a word? Or maybe just delete "of"?

Version 1:

Reviewer comments:

Reviewer #1

(Remarks to the Author)

I've gone through the revisions, and the authors have addressed my previous comments.

Reviewer #2

(Remarks to the Author)

I very much enjoyed reading this manuscript again. I would like to thank the authors for the level of detail and care they have given in responding to the reviews. The paper makes a significant and valuable contribution to the body of knowledge around coseismic deformation particularly in quantifying the near-fault deformation. I'm happy to recommend the paper for publication.

Reviewer #3

(Remarks to the Author)

My questions and concerns have been answered and addressed, and I still support publication. I think the authors have done a very thorough job responding to my and the other reviewers' comments and questions, and I thank them for their clear and detailed responses.

Open Access This Peer Review File is licensed under a Creative Commons Attribution 4.0 International License, which permits use, sharing, adaptation, distribution and reproduction in any medium or format, as long as you give appropriate credit to the original author(s) and the source, provide a link to the Creative Commons license, and indicate if changes were

made.

Reviewer #1

This manuscript focuses on the 2023 Kahramanmaraş earthquake sequence in Turkey, using InSAR geodesy to constrain the 3D displacement field spanning both the interseismic, coseismic, and early postseismic period. The authors examine the near-fault displacement pattern, and attempt to characterize the style of deformation around the fault surface rupture. They conclude that a significant proportion of deformation (~35%) is accommodated within a diffusely deforming damage zone, extending on average 5 km either side of the rupture, which generally increases in width close to geometric complexities (e.g. fault step-overs, etc). The main conclusion is that these off-fault damage zones may accommodate deformation over wider regions than typically considered in studies employing optical image correlation data, and that on-fault coseismic slip may not be representative of the total surface offset accommodated across the entire fault zone during an earthquake.

The paper is well-written, and contains much useful additional information in the supplementary material, although I did find it a bit complicated at times going back and forth between the main paper, the methods section, and the supplementary figures (which are important to the story in some cases). The topic is of interest, and the conclusions somewhat provocative, with potential interest for the community. However, I didn't always find the interpretations and discussion fully supported by the assumptions made (e.g. regarding ASD), or the data as presented; I have some reservations about some of the interpretation, and questions regarding potential artifacts arising from the 3D displacement inversion. This paper could be strengthened by demonstrating the 3D displacement result is not biased in the near-field (ideally by projecting the 3D displacement back into radar geometry and plotting the residuals in each InSAR dataset), as well as integrating with the optical data, which can give similarly good medium and near-field displacements. For me, the paper currently doesn't conclusively demonstrate that the near-field roll-over of displacement into the fault zone is the result of inelastic deformation within a very wide damage zone. Some of the results are also similar to existing studies looking at the fault slip distribution. But nevertheless, I found the paper interesting and easy to read.

Thank you for your recommendation and comments. Please see the point-by-point response below. In particular, we demonstrate the validation of our 3D displacements by comparing them with Sentinel-2 optical displacements from Ma et al. (2024). Please let us know if further revisions are needed to improve this manuscript.

Below, the comments are in **black**, our responses are in **red**, and the sentences modified in the manuscript are highlighted in **blue**. Figures supporting our perspective are labeled as **Figure R#** in the response letter.

Ma, Z., Li, C., Jiang, Y., Yun, S. H., & Wei, S. (2024). Space Geodetic Insights to the Dramatic Stress Rotation Induced by the February 2023 Turkey - Syria Earthquake Doublet. Geophysical Research Letters. <https://doi.org/10.1029/2023GL107788>

My main comments are below:

1. The authors do not utilise all the available geodetic data, ignoring optical correlation measurements which could significantly improve their 3D displacement dataset.

We did not utilize the optical results to calculate 3D displacements for the following two reasons:

(1) The Sentinel-2 optical displacements, captured shortly after the earthquake and shared on social media (https://x.com/Max_VWDV/status/1623809681641684993 and https://x.com/Geo_GIF/status/1624099723878731776), appeared noisy and exhibited systematic bias. Consequently, we did not plan to use the Sentinel-2 optical displacements when we initiated this work. COMET (The Centre for Observation and Modelling of Earthquakes, Volcanoes and Tectonics) combined SAR and Sentinel-2 optical displacements to calculate the 3D displacements of the earthquake. The following Figure R1 shows the comparison of horizontal displacements obtained in this paper with those obtained by COMET and the early Sentinel-2 optical displacements. The displacements obtained by COMET show significant residuals likely propagated from the Sentinel-2 optical displacements. Other optical images, such as WorldView and Pléiades, could also be used, but they are commercial images and are susceptible to cloud cover.

[figure redacted]

Figure R1. Comparison of horizontal displacements obtained from different institutes. Sentinel-2 displacements (from https://x.com/Max_VWDV/status/1623809681641684993) exhibit notable discrepancies when compared to our SAR-based results. COMET's results, derived from both SAR-based and Sentinel-2 displacements, also display significant residuals. The sources of these residuals are twofold: first, they originate from the Sentinel-2 displacements, as the residuals from COMET and Sentinel-2 show similar spatial patterns; second, they arise from the traditional weighted least squares method used by COMET to combine different displacement observations for 3D displacements. This method is less effective at suppressing high-frequency noise and accurately determining the weight of different observations

compared to the SM-VCE method used in this manuscript. EW, east-west displacement; NS, north-south displacement. (unit: meter)

(2) Afterward, a new version of Sentinel-2 optical displacements was calculated by Dr. Chenglong Li and Dr. Zhangfeng Ma from Nanyang Technological University (i.e., Ma et al. 2024, GRL). They applied de-ramping and de-striping corrections (Stumpf et al., 2018) to eliminate the systematic shifts and artifacts from the initial MicMac displacement fields of Sentinel-2 images. As shown in following Supplementary Fig. 6, the reliability of this version of Sentinel-2 displacements is improved significantly compared with those shown in Figure R1. However, as we had finished the 3D displacement calculation when this new version of Sentinel-2 displacement became available, we didn't incorporate this dataset into our calculation. Conversely, this new version of Sentinel-2 displacement can serve as an independent dataset to validate our SAR-based horizontal displacement. Also, the fault-parallel displacements along nine selected profiles show quite similar pattern between this Sentinel-2 result and our SAR-based results (Supplementary Fig. 6), further validating our SAR-based 3D displacements.

Stumpf, A., Michéa, D., & Malet, J. P. (2018). Improved co-registration of Sentinel-2 and Landsat-8 imagery for Earth surface motion measurements. Remote Sensing, 10(2), 1–20. <https://doi.org/10.3390/rs10020160>

For better demonstrating the validation of our 3D displacements, we added the following Supplementary Fig. 6 in the revised manuscript and mentioned in the main text as follows (Lines 116-118):

“Our SAR-based east-west and north-south displacements are consistent with displacements derived from Sentinel-2 optical images³⁶ in both the near-fault and far-field areas (Supplementary Fig. 6).”

Supplementary Fig. 6 The comparison of displacements obtained by Sentinel-2 optical images² and by SAR data in this paper. **a** and **b** are east-west (EW) and north-south (NS) displacements obtained from Sentinel-2 images. **c** and **d** are the residuals of **a** and **d** with respect to the SAR-based EW and NS displacements derived here (i.e., Supplementary Figs. 1a-b). The magenta lines with labeled number in **a** are selected profiles, and **e** shows the fault-parallel displacement comparison between the Sentinel-2

displacement and our SAR-based displacements. The insert panel in **c** is a zoom-in view of the dashed rectangle in **c**. It can be seen that our SAR-based horizontal displacements are well consistent with the Sentinel-2 optical results. Even if there is higher magnitude of residuals in the near-fault areas, the spatial extent is limited to only hundreds of meters, which has as negligible influence for our analysis of the up to 5 km width of the off-fault damage.

2. Neither do the authors demonstrate how well their 3D displacement result (which is obtained through an inversion procedure) fits the raw data.

The following figures, now Supplementary Figs. 7 and 8, show the pixel-offset tracking (POT) residuals within the 10-km buffer zone along the main ruptures. It can be seen that although there are obvious residuals in these near-fault POT observations, there are no systematic deviations, suggesting that our 3D displacement result within the near-fault regions fits well with the raw data. The residual maps of the standard differential InSAR (DInSAR), along-track multiple aperture interferometry (MAI), and range split-spectrum interferometry (RSSI) observations are not presented here since these observations are almost completely decorrelated in the near-fault areas. To better illuminate this point, these two residual maps have been added in the revised supplementary information and the following sentences added in the main text (Lines 118-121):

“By projecting the 3D displacements back into the line-of-sight SAR geometry, we obtain residuals of the original observations (Supplementary Figs. 7 and 8). These residuals show no systematic deviations, indicating that no single input data set is leading to a bias in the 3D displacement derivation.”

Supplementary Fig. 7. **a** Pixel-offset tracking (POT) residuals within a 10-km buffer zone along the ruptures of the Mw7.8 event. These residuals are obtained by differencing between the original observations and projected observations from the 3D displacements. **b** Residual values of the different observation sets along the profile A-A' in **a**, where the thin colored lines represent the corresponding observations in **a** with the same colored boundary, and the thick black line is the average value of the colored lines. The location of $x=0$ in **b** represent the location of the fault. Although there are obvious residuals in these near-fault POT observations, there are no systematic deviations, suggesting that no single input data set is leading to a bias in the 3D displacement derivation. The residual maps of the standard differential InSAR (DInSAR), along-track multiple aperture interferometry (MAI), and range split-spectrum interferometry (RSSI) observations are not presented here since these observations are almost completely decorrelated in the near-fault regions. S1, Sentinel-1; As, ascending; Des, descending; T, track; azi, azimuth; rg, range.

a Residual along ruptures of the Mw7.6 event

Supplementary Fig. 8. Same as Supplementary Fig. 7 but for the ruptures of the Mw7.6 event.

3. The methodology used to obtain the displacement is not discussed in much detail (referring the reader to another more technical paper in IEEE TGRS). However, the two papers cited to justify this 3D inversion technique (Liu, et al., 2018, 2022) both integrate InSAR data which extends close to the fault rupture, and therefore do not rely on the much noisier pixel offset tracking data to constrain the nearfield. However, in this Turkey earthquake sequence, the near-field offsets are largely constrained by pixel offset tracking, since the InSAR data does not extend anywhere near the rupture for Sentinel-1, and is even quite far for ALOS-2.

Thank you for your comments. There is no difference in calculating 3D displacements whether the observations are from InSAR data or pixel offset tracking (POT) data, as long as the observations are valid within the study area. Various case studies, such as those focused on the coseismic 3D displacements based primarily on POT observations within near-fault regions (e.g., the 2021 Maduo (China), 2019 Ridgecrest (CA), and 2016 Kaikoura (New Zealand) earthquakes), demonstrate the capability of our SM-VCE method to reliably calculate 3D displacements from noisier POT observations. Furthermore, employing up to 12 POT-based observations to calculate the 3D displacements (i.e., three unknowns) within near-fault regions can provide sufficient data to enhance the reliability of SM-VCE-obtained 3D displacements. Also, Supplementary Figs. 7-8 suggest that the 3D displacements calculated by our SM-VCE method fit well with the original POT displacement observations.

To better clarify the capability of our SM-VCE method in this case study, we added the following sentences in the revised manuscript (Lines 387-392).

“Even if only the noisier POT observations are available in the near-fault areas, the 3D displacements can still be well resolved, e.g., for the 2021 Maduo (China)⁶⁷ and 2016 Kaikoura (New Zealand)⁶⁸ earthquakes. Here we combined twenty-three independent SAR displacement observations (Supplementary Fig. 4) to calculate the 3D displacements with the SM-VCE method. In the near-fault areas, up to 12 independent POT-based observations are available, providing sufficient data for reliable 3D displacement derivation.”

4. The authors do not give many details about the unwrapping technique they use, which is probably less robust than, say, the branch cut method, which may be better for dealing with discontinuities.

We use the minimum cost flow (MCF) method to unwrap interferograms. For the coseismic interferograms, the coherence of most near-fault regions is too low to obtain a reliable unwrapping result due to the extremely dense fringe and the ground surface change. Therefore, we manually masked out the low-coherence area before MCF unwrapping, and iteratively checked the unwrapping result to eliminate possible unwrapping errors. This unwrapping process has been illuminated in the Methods—Coseismic SAR data processing section in the revised manuscript (Lines 335-336).

“This was followed by minimum cost flow interferogram unwrapping after manually and iteratively masking out low coherence areas”

5. Given the above comments, it remains unclear what bias is introduced to the near-field displacement results using their inversion procedure, which is heavily weighted towards the extremely noisy pixel offset tracking results. The inversion procedure relies on some form of regularization, I don't know to what extent this impacts the near-field displacements? This, for me, may be critical, because the scientific novelty of this paper is focused on the near-field displacement data.

Based on our above responses and information, we think our 3D displacements are unbiased and would not affect the discussion and conclusion of the extensive 5-km-width off-fault damage zone. Yes, we agree that our 3D displacements look smooth, but we didn't add any mathematical form of regularization during the calculation. This smoothness comes from the employment of the strain model in SM-VCE method, which, in simple term, can be considered as a low-pass filter since we assume the deformations within a window are correlated. As shown in the following, the strain model assumes that the deformation gradient within this window is a constant and the deformation value changes smoothly within this window. This assumption is reasonable since we adopt a window with the size of $750\text{ m} \times 750\text{ m}$ in this manuscript which is much smaller than the 5 km of the off-fault damage zone, and it is usually the case to conduct the low-pass filter within this extent for SAR-based displacements.

For better clarifying this point, the following Supplementary Fig. 25 and sentences (Lines 399-404) have been added to the revised manuscript.

“Note that although our SM-VCE-obtained coseismic 3D surface displacements look smooth, this smoothness comes more from the employment of the strain model in the SM-VCE method, rather than adding any mathematical form of regularization during the calculation. The rationale of the strain model lies in that the deformation gradient within a window (i.e., $750\text{ m} \times 750\text{ m}$ in this paper) is a constant and the deformation value changes smoothly within this window (Supplementary Fig. 25).”

Supplementary Fig. 25 A schematic diagram of the strain model illustrates how the deformation of adjacent points is described to enhance understanding of the SM-VCE (Strain Model and Variance Component Estimation) method⁵ used in this study to obtain coseismic three-dimensional (3D) surface displacements. Generally, deformation at different points within a window is correlated, and the strain model assumes a constant deformation gradient within this area. In this approach, SAR observations (blue pixels) around a target point (red pixel) are used to estimate the 3D displacement components of the target, providing a more robust result than the standard weighted least squares method^{6,7}, which only incorporates SAR observations at the target point itself.

6. The correlation window also uses relatively small window sizes for the range direction (ALOS-2) and azimuth direction (Sentinel-1), which may in fact be too small to obtain robust displacements in these components; when the window size is too small, there is not enough information to obtain a highly accurate result, thus imparting a sub-pixel bias which tends toward zero; this effect is commonly seen with optical correlation data when using windows too small given the level of noise in the image windows (unless mitigated by spatial regularization, but this can also smooth out discontinuities). Given that amplitude images are relatively noisy at high frequencies (due to the speckle), it's unclear if 32 pixels is appropriate for resolving an unbiased displacement in the range direction (ALOS-2) and azimuth direction (Sentinel-1).

We agree with you that the small window size of 32 pixels used for the pixel correlation will reduce the accuracy of the obtained displacements. However, too large window sizes will smooth the displacement values within the near-fault regions. The following Supplementary Fig. 21 shows the S1_DesT21_POT displacements estimated with different window sizes. In general, the displacement from different window sizes results in similar displacement pattern without any significant deviations, and the result of larger window of 512*128 is smoother than that of the small window of 128*32. Nevertheless, the most seriously affected region with large correlation windows is in the near-fault areas. As shown in the insert zoom-in view of Supplementary Fig. 21, the large window smooths out the sharp displacement offset across the fault, which will significantly affect the following near-fault off-fault damage analysis. We also obtained the ALOS2_DesT77_POT displacements with different window sizes (Supplementary Fig. 22), and the difference from the Sentinel-1 data lies that the near-fault displacement offset is not seriously smoothed out by the larger window size for the ALOS-2 data. This difference may result from the different imaging processes for ALOS-2 (ScanSAR) and Sentinel-1 (TOPS) data. Nevertheless, there is smoothness across the fault with larger window size (the insert maps in Supplementary Fig. 21b) for the ALOS-2 data. Therefore, we prefer to use the small correlation window for the pixel-offset tracking process to reserve the displacement details as much as possible. Although the result of small window contains higher level of high-frequency noise, the signal-to-noise ratio of the 3D displacement is improved around the near-fault regions by using the SM-VCE method with the assistance of manually selected fault trace (see Methods).

For better clarifying this point, we added the following Supplementary Figs. 21 and 22 and sentences to the revised manuscript (Lines 346-349).

“We did not use larger coregistration windows for the POT processing since larger windows could smoothen out the sharp displacement offset across the fault and bias our off-fault damage analysis (Supplementary Figs. 21 and 22).”

Supplementary Fig. 21 Pixel-offset tracking (POT) displacements estimated with different window sizes, i.e., 128*32 and 512*128 (range*azimuth), for descending (Des) Sentinel-1 (S1) track 21 (T21) images. azi, azimuth; rg, range. It can be seen that the main difference between different window sizes is the smoothness

of the derived displacement, i.e., larger windows smoothen high-frequency noise. However, in the near-fault areas, the large window also smoothen the sharp displacement offset across the fault, which can bias our off-fault damage analysis. Therefore, we prefer to use the smaller correlation window for the pixel-offset tracking process to preserve the displacement details as much as possible.

Supplementary Fig. 22 Same as Supplementary Fig. 21, but for descending (Des) ALOS2 track 77 (T77) images.

7. The fault-parallel displacement profiles in Fig. 2 appear highly smooth in the near-field region. I compared these with results from correlation of Sentinel-2 data, and they look overly smooth in the near-field region. Also, the residuals (once the model results have been removed) show spatial variability with length-scales of ~ 5 km, which suggests maybe some level of smoothing is present in this displacement data (presumably resulting from the 3D inversion method?).

The smoothness of our 3D displacement comes from the employment of our SM-VCE method that utilize the strain model to establish the Green function between SAR-based observations and 3D displacement (the above response #5). We compared our 3D displacements with the Sentinel-2 result, and they are consistent with each other (the above response #1).

As for the ~ 5 km length scale of residuals in Fig. 2, we attributed it to the topography-related signal (see the following Supplementary Fig. 2). This is reasonable since the accuracy of the pixel-offset tracking (POT) displacement is generally affected by the local topography (Liu et al. 2020). For better clarifying this point, Supplementary Fig. 2 and the following sentences have been added in the revised manuscript (Lines 91-93).

“In the residuals, there is a signal with a length scale of ~ 5 km, which can be attributed to topography-related artifacts from the pixel-offset tracking (POT) process of non-orthorectified SAR images (Supplementary Fig. 2).”

Liu, X., Zhao, C., Zhang, Q., Lu, Z., & Li, Z. (2020). Deformation of the Baige Landslide , Tibet , China , Revealed Through the Integration of Cross - Platform ALOS / PALSAR - 1 and ALOS / PALSAR

Supplementary Fig. 2. Same as Fig. 2d in the main text, but with additional elevation profile, suggesting that the ~ 5 km length scale of residuals (yellow) at around -20 km could be attributed to topography-related signals. This is reasonable since the accuracy of the pixel-offset tracking (POT) displacement from non-orthorectified SAR images is generally affected by the local topography¹.

8. Since these near-field displacement measurements are critical for the main story of this paper (that damage zone widths extend further than previously thought), I would encourage the authors to perhaps (1) supplement their results with those from optical correlation data, which provides an independent and more detailed view of the near-field displacement), and (2) to see how well their 3D displacement solution accounts for the information in their raw displacement data; for example, projecting the 3D displacements back into radar geometry and plotting the residuals would be important to convince the reader that the near-field displacements are well resolved. Such an endeavor should also show appropriately zoomed in views of the near-field region, as it would be challenging to see at the scale shown in Fig. S3.

Thanks for your comments, and please see the responses to comments #1 and #2.

9. I found the section on calculating ASD slightly confusing. I follow the concept that a long-term block-like offset minus the interseismic can give you the co-seismic (plus postseismic, and any potential interseismic creep bursts, which are somewhat ignored here). However, the interseismic observations do not cover the entire interseismic period, therefore how do you know what the total displacement should be, i.e. the magnitude of the long-term offset (the step in Fig 2a)? Maybe you compute the long-term step over the same time-period as the interseismic observations? But this would ignore any temporal variations in the interseismic strain accumulation, which could be possible. Also, this approach assumes the coseismic has a uniform down-dip displacement, which you demonstrate later with your subsequent slip inversion (Fig. 3b) not to be the case.

Figure 2a-2c in the main text

Figures 2a-2b are presented to illuminate the ideal displacement pattern (as well as the mathematical expression) during one full earthquake cycle and the interseismic period, respectively, based on which we can obtain the ideal displacement pattern and the mathematical expression of coseismic deformation.

$$C(x) = L(x) - I(x) = \begin{cases} -0.5s_0 - \frac{s_0}{\pi} \cdot \arctan\left(\frac{x}{D}\right) & x < 0 \\ 0.5s_0 - \frac{s_0}{\pi} \cdot \arctan\left(\frac{x}{D}\right) & x > 0 \end{cases}$$

The ASD is obtained as the residual between the coseismic displacement observations with respect to the predicted elastic coseismic displacement, which is obtained by fitting equation $C(x)$ to the coseismic displacement observations. During this fitting process, the long-term offset s_0 and the locking depth D are taken as unknowns, therefore it is unnecessary to determine the step s_0 in Fig 2a in advance.

Yes, the implicit assumption is the uniform coseismic slip, which is generally not the case in reality. Nevertheless, our fitting results of profiles along the earthquake ruptures (see the supplementary movie) indicate that $C(x)$ can well describe the observations, demonstrating the validation of this assumption.

To better illuminate this point, we added the following sentences in the revised manuscript (Lines 423-429):

“Note that it is not necessary to determine the relative offset s_0 for equation (1) and the locking depth D for equation (2) in advance; these two parameters are taken as unknowns in equation (3) and are estimated when fitting equation (3) to the coseismic displacements. Although the implicit assumption of equation (3) is uniform coseismic slip within the seismogenic depth, our Supplementary Movie demonstrates the capability of equation (3) in fitting the coseismic displacements, indicating a negligible effect of this assumption on the analysis conducted here.”

10. The profile chosen in Fig. 2d is maybe not a good/representative case, because it is located where the Narli fault intersects the main EAF. Therefore, you have chosen a place where the fault structure is more complicated, and thus not well represented by the simple block model shown in Fig. 2a. Indeed, the locations where the fault zone width appears relatively high correlate with locations of structural complexity, and thus perhaps simply a failure of your simple block model to accurately reflect the complexity in the near-field structure.

We agree with you that it would have been better to choose a profile around simpler sections of the fault. However, Figure 2d is shown as a schematic diagram, more to illustrate the calculation of the absent surface displacement (ASD) and the meaning of on-fault displacement, total displacement, and ASD. Besides, from the fitting results of profiles along the earthquake ruptures (see the Supplementary Movie), we can say that the displacement pattern shown in Fig. 2d is a fairly general representative of the fault-parallel displacement with absent surface

displacement (ASD) in both fault complexity sections and simple sections along the fault. Therefore, we prefer to keep the location of Fig. 2d unchanged. To better illuminate this point, we added the following sentences in the revised manuscript (Lines 95-99):

“The grey bar in Fig. 2e (within the Fig. 4d rectangle) represents the center position of the profile (grey dots) shown in panel **d**, which we picked because its displacement pattern is fairly typical of the fault-parallel displacements with ASD in both complex and simple sections along the fault (see Supplementary Movie).”

Our “block model”, as illuminated in the Supplementary Movie, is capable of depicting the displacement observations outside the damage zone. While within the damage zone, especially in the structurally complex regions, more serious inelastic deformations occurred during the earthquake. The absent surface displacement (ASD) within these regions is one geodetic manifestation that could be observed by InSAR as shown in this manuscript, where the extent and magnitude of the ASD can be quantitatively assessed. However, it is difficult to accurately determine the internal structure details within the damage zone based only on the InSAR displacement observations. Future investigations combining with other observations (e.g., Ground Penetrating Radar) are required to shed light on the near-field structure details. To better illuminate this point, we added the following sentences in the revised manuscript (Lines 447-453):

“By fitting the fault-parallel displacements along fault-perpendicular profiles with equations (3) and (4), we can obtain the magnitude and extent of the off-fault damage. However, it is difficult to accurately determine the internal structure details within the damage zone based only on the SAR-based observations since the observed ASD within these regions is just one geodetic manifestation that could be captured by SAR images. Future investigations combining with other observations (e.g., ground penetrating radar) are required to shed light on the near-field structure details.”

11. A more minor point, but the variation of the fault strike over the width of the swath profile used to stack the fault-parallel displacement profiles can also artificially widen the apparent fault zone width (see Milliner, et al. 2021) (although I acknowledge probably not to the extent of 5 km).

We agree with you. However, since our 3D displacements are smooth after applying the SM-VCE method inversion, we did not perform stacking within a swath to increase the signal-to-noise ratio of the final fault-parallel displacement profile. Instead, after determining the location of a profile line, we select points along this profile at 50 m intervals (matching the 50 m spatial resolution of the 3D displacements). We then calculate the fault-parallel displacements at these points by averaging the nearest four neighbors, excluding the neighbor on the opposite side of the fault. To better illustrate the process of determining the fault-parallel displacement along fault-perpendicular profiles, we have added the following sentences to the Methods section in the revised manuscript (Lines 439-446).

“The above calculations are based on fault-parallel displacements of each fault-perpendicular profile. Each fault-perpendicular profile is oriented perpendicular to the local fault strike and extends 50 km away from the fault on each side. We select points along each profile every 50 meters (given the spatial resolution of the 3D displacements is ~50 m) and calculate the displacement values at these points by averaging coseismic east and north displacements of the nearest four neighbors, excluding neighbors on the opposite side of the fault. Finally, the fault-parallel displacement is obtained by projecting the averaged horizontal displacements onto the profile.”

Milliner, C., Donnellan, A., Aati, S., Avouac, J. P., Zinke, R., Dolan, J. F., Wang, K., & Bürgmann, R. (2021). *Bookshelf Kinematics and the Effect of Dilatation on Fault Zone Inelastic Deformation: Examples From Optical Image Correlation Measurements of the 2019 Ridgecrest Earthquake Sequence*. *Journal of Geophysical Research: Solid Earth*, 126(3), 1–20. <https://doi.org/10.1029/2020JB020551>

12. The similarity in the ASD and SSD plots in Fig 3d looks to me like the ASD is simply showing the same thing as SSD. In the case of SSD, we're looking at the difference between peak slip (around 5 km depth) and the surface. Since you assume a simple discontinuity with no constraints shallower than 5 km, it seems unsurprising that you get the same result.

We are sorry that we don't quite understand your comment that "you assume a simple discontinuity with no constraints shallower than 5 km". Do you mean that we didn't add the smooth constraint for patches shallower than 5 km for the kinematic slip model? Actually, we treat all the patches equally, and assume that the slip on a target patch to be similar to the inverse-distance weighted average slip of the surrounding patches. This statement can be found in the Methods---Finite-fault inversion Section (Lines 485-490).

Please let us know if we misunderstood your comment.

"Since patches are unevenly distributed on the fault plane, we imposed inverse distance-weighted smoothing constraints between adjacent patches, which means that the inversion procedure favors slip on a target patch to be similar to the inverse-distance weighted average slip of the surrounding patches (Supplementary Fig. 27). A smoothing factor of 10 was selected based on a trade-off curve between the normalized slip roughness and observation misfit (Supplementary Fig. 28)."

13. Regarding the apparent SSD resolved in your slip model is that maybe this is simply a feature of the data, and that the reduced slip in the shallow crust is well-resolved, and there is no distributed off-fault damage at all? In other words, why does this slip deficit have to be distributed off-fault damage, and not simply tapered slip towards the surface?

We absolutely agree with you that we cannot say the modelled SSD is coming from the real tapered slip or from the off-fault damage based only on the modeling of coseismic displacements with an elastic model.

To distinguish if the tapered slip is a real feature, we discussed in the beginning of the Discussion section of the manuscript. We first assume that the tapered slip is a real feature, in which case the missing coseismic shallow slip should be compensated during interseismic slip or postseismic afterslip given that the total slip accommodated by the fault, over the entire earthquake cycle, should be the same from the surface down to the fault locking depth. However, the InSAR 2014-2019 interseismic and the first nine months of postseismic displacements indicate almost no surface discontinuities, therefore rule out the complementary shallow slip during the InSAR observing period (2014-2019, 2023.2-2023.11). Conversely, there is ample evidence for off-fault damage being responsible for the modelled SSD, i.e., distributed surface ruptures and fringe discontinuities in coseismic L-band ALOS-2 interferograms. Therefore, analysis of the interseismic/postseismic displacements and the coseismic interferograms support the responsibility of off-fault damage, rather a real SSD, on the modelled tapered slip towards the surface.

Since the interseismic displacement is only available between 2014-2019, we cannot completely rule out the possibility of shallow interseismic slip (e.g., aseismic creep and transient slow-slip event) during other parts of the interseismic period (i.e., before 2014). However, the moment of the estimated SSD, needed to compensate for the observed coseismic

SSD, is large at 5.65×10^{19} Nm, corresponding to a Mw 7.1 event. Even if transient slow-slip events occurred outside the InSAR observation period, their magnitude would typically be only a few millimeters, with recurrence intervals of several years^{42,43}. It is thus unlikely that transient slow-slip events compensated for the SSD of 2-4 meters during historical times without being detected. Therefore, we would like to support the point that the majority of the modelled coseismic SSD is related to the off-fault damage, rather than to real SSD.

Details about this point can be found in the beginning of the Discussion section in the manuscript (Lines 214-255).

14. In the study of Marchandon, et al. (2021), the authors demonstrate that inverting with an overly simplistic model can artificially enhance slip at depths of ~5 km. Therefore, even if the slip at the surface is well-resolved (e.g. by pixel tracking or optical correlation data), one may infer an apparent SSD from the slip model because the slip at depth is enhanced. Could that be the case here, since the slip inversion is made with a homogenous elastic half space with a somewhat simple fault model (given the complexity we see along the fault rupture), etc? In this case, SSD is actually the result of overestimating the slip at depth, rather than underestimating the slip at the surface?

Yes, we agree with you that the simplistic homogenous elastic model would bias the slip estimation at depth when underground elastic properties are heterogenous. Nevertheless, this overestimation of slip at depth has negligible effect on the correlation analysis between the absent surface displacement (ASD) and SSD ratios in Figs. 3c-d in this paper.

To better illuminate this point, we added Supplementary Text 1, Supplementary Fig. 14, and the following sentences in the revised manuscript (Lines 195-198):

“Supplementary Text 1

Do geodetic slip inversions using simple homogeneous elastic half-space models significantly under- or overestimate shallow slip deficit?

Marchandon et al. (2021)³ demonstrated that when inverting for fault slip with a simple homogenous elastic half-space model, it can artificially enhance the slip at depths of ~5 km. In this case, even if fault slip close to the surface is well-resolved by geodetic observations (i.e., SAR-based displacements in this paper), one may infer an apparent shallow slip deficit (SSD) from the slip model because the slip at depth is overestimated. Nevertheless, this overestimation of slip at depth has negligible effect on the correlation analysis between the absent surface displacement (ASD) and SSD ratios in Figs. 3c-d in the main paper. As demonstrated by the simulated experiments in Marchandon et al. (2021), the simplification of heterogenous underground elastic properties with a homogenous model would result in the overestimation of slip magnitude about 0.74 m between 3-6 km, which corresponds to about 20% of the input slip model at this depth. Here, we recalculated the SSD ratio for the 2023 Kahramanmaraş (Türkiye) earthquakes by artificially subtracting 20% from the maximum slip at depth of our slip model, aiming to account for the effect of the homogenous elastic model in overestimating the maximum slip at ~5 km depth. As shown in Supplementary Fig. 14, the SSD ratios are highly correlated ($R=0.969$) before and after subtracting an artificial slip value from the maximum slip value at depth. Even if we subtract a larger proportion of the maximum slip (e.g., 30%), the correlation between two sets of SSD ratios is still very high. Besides of the SSD ratio, in the Fig. 3 of Marchandon et al. (2021), we can find that the SSD depth (i.e., the depth of maximum slip) seems unchanged with different underground elastic models. Therefore, we conclude here that using a homogenous elastic model has negligible effect on

the estimation of the SSD ratio and on the correlation analysis between ASD and SSD ratios in this paper (i.e., Fig. 3c-d in the main text).”

Supplementary Fig. 14 Comparison of the shallow slip deficit (SSD) ratio under the case that the maximum slip ($Slip_{max}$) value at depth is overestimated by 20% (blue) and 30% (red) due to using a simple elastic half-space model in the slip inversion³. In the legend, R represents the correlation coefficient. The black line is $y=x$. It can be seen that the SSD ratio values are highly correlated before and after correcting for a possible overestimation of the maximum slip, indicating that our simple homogenous elastic half-space model has little effect on the correlation analysis between the ASD and SSD ratios in this paper (i.e., Figs. 3c-d in the main manuscript, see Supplementary Text 1 for details).

“Although it is possible to overestimate slip values by $\sim 20\%$ at depths of ~ 5 km when using a simple homogenous elastic inversion model⁴⁰, this overestimation has a negligible effect on the correlation analysis between the ASD and SSD ratios (see Supplementary Text 1 and Supplementary Fig. 14).”

[figure redacted]

Fig. 3 in Marchandon, et al. (2021). Normalized cumulative slip with depth of the input slip model and the three retrieved slip models. The shaded area of each curve represents the uncertainty due to the smoothing of the distribution of cumulative slip with depth. The amount of SSD is indicated at the top of each curve.

This figure shows that the SSD depth (i.e., the depth of maximum slip) seems unchanged with different underground elastic models.

Marchandon, M., Hollingsworth, J., & Radiguet, M. (2021). Origin of the shallow slip deficit on a strike slip fault: Influence of elastic structure, topography, data coverage, and noise. Earth and Planetary Science Letters, 554, 116696. <https://doi.org/10.1016/j.epsl.2020.116696>

15. How do you decimate your displacement data for use in the inversion? Or rather, how do you avoid smoothing over the fault with your minimum quadtree block size, while retaining enough near-field data to adequately constrain the shallow-most slip?

We take the surface fault trace as additional input when downsampling the displacement data, which helps to exclude the points on the other side of the fault for blocks close to the fault trace. We added the following sentences to clarify this point in the revised manuscript (Lines 490-494):

“Using this setup, quadtree-downsampled 3D displacements (Supplementary Fig. 29) were used to invert for the fault slip distribution in a uniform elastic half-space¹², in which the surface fault trace is taken as additional input to exclude the points on the other side of the fault for quadtree blocks close to the fault trace.”

16. Field measurements of fault damage zones are generally rather narrow (e.g. Mitchell, et al. 2009), much less than 5 km as reported here. Similar observations for dense seismic experiments across fault zones show similar things (Li, et al., 2007). How do you reconcile these observations with your interpretation of such a large damage zone? Also, how do you interpret these damage zones in the context of healing, as seen in Vp/Vs studies (Brennguier, et al., 2008)?

The off-fault damage may appear as distributed deformation (e.g., warping and rigid-block rotation), which may not cause obvious surface marks (e.g., ruptures) or produce significant rock failure within the bulk medium, thus it would be hard to accurately identify the extent of off-fault damage zone by field investigation or seismic wave velocity estimation.

For the healing of the coseismic damage zones, as revealed by the increased seismic velocity around the fault during the postseismic phase, it is usually assumed that cracks within the damage zone opened during the main shock and gradually closed afterwards (e.g., Li et al., 2003, Vidale et al., 2003). This healing process geometrically makes the medium in the damage zone look similar as the bulk medium (i.e., the geometric closure of the opened cracks), but its physical properties may have changed, making it vulnerable to damage during the next earthquake. In our opinion, this kind of healing may be attributed to the internal force from the bulk medium, e.g., the poroelastic pressure and the overall tectonic movement. Nevertheless, this manuscript provides a geodetic view about the coseismic off-fault damage, and it would be very interesting to combine geodetic and seismic datasets during both coseismic and postseismic analysis in future studies.

Li, Y. G., Vidale, J. E., Day, S. M., Oglesby, D. D., & Cochran, E. (2003). Postseismic fault healing on the rupture zone of the 1999 M 7.1 Hector Mine, California, earthquake. *Bulletin of the Seismological Society of America*, 93(2), 854–869. <https://doi.org/10.1785/0120020131>

Li, H., Zhu, L., & Yang, H. (2007). High-resolution structures of the Landers fault zone inferred from aftershock waveform data. *Geophysical Journal International*, 171(3), 1295–1307. <https://doi.org/10.1111/j.1365-246X.2007.03608.x>

Mitchell, T. M., & Faulkner, D. R. (2009). The nature and origin of off-fault damage surrounding strike-slip fault zones with a wide range of displacements: A field study from the Atacama fault system, northern Chile. *Journal of Structural Geology*, 31(8), 802–816. <https://doi.org/10.1016/j.jsg.2009.05.002>

Vidale, J. E., & Li, Y. G. (2003). Damage to the shallow Landers fault from the nearby Hector Mine earthquake. *Nature*, 421(6922), 524–526. <https://doi.org/10.1038/nature01354>

Brenguier, F., & Campillo, M. (2008). Postseismic Relaxation Along the San Andreas Fault at Parkfield from Continuous Seismological Observations. *Science*, 321(September), 1478–1481. <https://doi.org/10.1126/science.1160943>

We added the following sentences to clarify this point in the revised manuscript (Lines 296-308):

“Field investigations and seismic experiments also show relatively narrow zones of off-fault damage^{20,54}, typically much less than the ~5 km reported here. The main reason for this discrepancy is that off-fault damage may manifest as inelastic rock deformation (e.g., warping and rigid-block rotation), which might not create obvious surface features (e.g., ruptures) or lead to significant rock failure within the bulk medium. Consequently, accurately identifying the extent of the off-fault damage zone through field investigations or seismic wave velocity estimation is challenging. Additionally, it has been shown that coseismic damage zones undergo healing process afterward, as indicated by the increased seismic velocities around the fault during the postseismic phase⁵⁵⁻⁵⁷. Hence, following this healing process, the medium within the damage zone is similar to the bulk medium (i.e., after the geometric closure of opened cracks⁵⁷). Here, we offer a geodetic perspective on coseismic off-fault damage, which should be integrated in geodetic and seismic datasets in both coseismic and postseismic analyses for future earthquakes.”

17. In terms of the conclusions, a field geologist would typically be aware of the problems associated with estimating a representative fault slip rate using Quaternary dating of offset geomorphic markers for a structurally complex segment of a fault (vs a structurally simple one). So it's not entirely simple to present this issue in such black and white terms. Ideally, one would estimate a fault slip rate on a simple segment, subject to less distributed off-fault deformation (though it really depends where the offset geomorphic markers can be found). It's maybe more true that trenching studies might be more biased towards structurally complex sections, where

vertical movements influence drainage networks, thus facilitating more continuous sedimentation.

We agree with your points. Geologists recognize the importance of selecting structurally simple sections of a fault to accurately estimate the slip rate. However, in practice, the availability of offset geomorphic markers is often limited. It is crucial to remember that off-fault damage can account for up to a third of the on-fault slip at depth, whether in geomorphic offset estimations or drainage trenching. Considering this, we conclude that our findings suggest the need for a broader reconsideration of slip-rate studies and seismic hazard assessments along major faults, rather than focusing solely on a specific aspect of fault slip rate studies. We added the following sentences to clarify this point in the revised manuscript (Lines 315-319):

“Still, due to this fault damage, near-fault slip-rate estimates could be underestimated by 10-20% on continental strike-slip faults and possibly as much as a third. This should also be accounted for in paleoseismological works where it is not always possible to find an ideal site near a structurally simple segment with limited off-fault damage.”

18. Line 241 claims that optical is unable to resolve the medium or far field displacement, but this is not the case. Jolivet, et al. (2013), and various other studies since, have shown very well that optical data can be used to capture the attenuation of displacement far from the fault rupture. This might be true with very high resolution optical data, but even Pleiades has a footprint that spans 15-20 km (depending on the incidence), and for an east-west striking fault, these images may capture the attenuation of slip over even greater distances.

Thank you for your comments. If we understand you correctly, you point to Line 251 (not Line 241) in the original manuscript. Our original intention was to convey that previous studies using optical data for off-fault damage analysis only utilized data from a few kilometers within the near-fault regions, as indicated in the "Data range used for analyzing damage" column of Supplementary Table 1. To clarify this point, we have revised the original sentences as follows (Lines 278-286):

“This is likely due to the narrower aperture of the displacement-field used in fault damage analysis of previous studies (i.e., hundreds of meters to several kilometers, Supplementary Table 1) compared to the 100-km-wide SAR-based displacement field used here. Calculating ASD involves extrapolating observations to determine the total deformation. Therefore, previous cases with smaller aperture of displacement are only sensitive to ASD at spatial scales much smaller than the width of the input dataset. This limitation prevents them from precisely deciphering the total far-field elastic deformation curve across the fault (i.e., the red model curve in **Fig. 2d**), even though the optical images are able to reveal the entire width of deformation pattern of earthquakes⁵⁰⁻⁵².”

Jolivet, R., Duputel, Z., Riel, B., Simons, M., Rivera, L., Minson, S. E., Zhang, H., Aivazis, M. A. G., Ayoub, F., Leprince, S., Samsonov, S., Motagh, M., & Fielding, E. J. (2014). The 2013 Mw 7.7 balochistan earthquake: Seismic potential of an accretionary wedge. Bulletin of the Seismological Society of America, 104(2), 1020–1030. <https://doi.org/10.1785/0120130313>

19. Line 255. I think these widths of 5 km more likely tell us the limitations of the fault slip inversion at capturing the precise details of the fault rupture within the upper 5 km.

If we understand you correctly, you mean that the fault geometry within the upper 5 km is not precise during the fault slip inversion, given that the ASD width is up to 5 km. In our view, assuming a planar fault geometry, the fault geometry within the upper 5 km is likely accurate enough. This is because we can clearly observe the surface displacement offset across the fault

rupture, indicating that the fault geometry at a depth of 0 km is correct. Additionally, the fault geometry below 5 km can be considered reliable, as the modeled surface displacement is consistent with our observations using the fault slip model. Therefore, the main fault plane within the upper 5 km used in the fault slip inversion is well constrained. However, the ASD width of up to 5 km indicates that the near-fault zone is vulnerable to off-fault damage, such as micro-ruptures. These secondary micro-ruptures are not detectable and are not included during the fault slip inversion.

We added the following sentences to clarify this point in the revised manuscript (Lines 477-485):

“We assume a planar fault geometry for each fault segment, where the topmost trace of the fault plane is fixed to the main fault trace, as determined by sharp offsets in the near-field POT results. The best dip angle of each fault segment was determined by a grid-search strategy (Supplementary Fig. 27). Although there may be small secondary fault ruptures within the ~5 km wide damage zone, apart from the main rupture, these are difficult to detect and incorporate into the slip model. By using planar fault segments aligned with the surface fault trace, the modeled surface displacements match well with our observations, indicating that the main fault plane within the seismogenic depth used in the fault slip inversion is well constrained.”

20. Fig. S4. What are the 2 arrow scales in (a)? Where is EKZ1 in (b)? This has a larger offset, and is closer to the fault, therefore more interesting than the very far-field measurements.

We have labeled the arrow scales as horizontal displacements and labeled EKZ1 in (b) as follows.

21. Fig. S11. This figure highlights how noisy the pixel offset tracking results are close to the rupture. It seems surprising to see how noisy these data are here, and how smooth the final inversion results are in Fig. 2 in the main paper, given that the pixel offset tracking data provide the main constraints on the near-field displacements.

See the above responses to comments #1 and #2.

22. Fig. S15. This idea has already been discussed by, for example, Scott, et al. (2018) and Miliner, et al. (2016). Also, case (a) is typically what is seen over very short length scales, while case (b) is seen even with optical data over length scales of several km.

In Scott, et al. (2018), Miliner, et al. (2016), and other related cases as listed in Table S1, they generally focused on the near-fault several kilometers (or even only hundreds of meters) displacements when analyzing the absent surface displacement (ASD), which is typically the case shown in Fig. S15a (attached in the following). The following Figure R2 is from Miliner, et al. (2016), where only up to 2 km of data is used for analyzing the ASD with a quite simple linear regression method. Nevertheless, we agree with you that the profile shape shown in Fig. S15c can also be seen from optical data, but previous studies only focused on the near-fault hundreds of meters to several kilometers regions, which prevents us capturing the large-scale signal of off-fault damage. Besides, unlike in previous studies, we employ the arctangent function of the coseismic displacements to fit the displacement outside the damage zone.

We added the following sentences to clarify this point in the revised manuscript:

“Previous studies assumed that the surface displacement decreases within the damage zone, consistent with our assumptions. However, they considered the displacement values outside the damage zone to be constant or vary linearly with the distance (Supplementary Fig. 26a), whereas we use the elastic arctangent function to fit the coseismic displacement outside the damage zone.” (Lines 458-462)

“Furthermore, while previous studies analyzed off-fault damage using near-fault observations limited to hundreds of meters to a few kilometers from the fault (Supplementary Table 1), our study uses 100-km-long displacement profiles, allowing us to capture the large-scale signal of off-fault damage.” (Lines 466-470)

[figure redacted]

Figure R2. Figure S7 in Miliner, et al. (2016) shows the displacement of several profiles, where black lines are observations and red lines are the fitting of black lines between the vertical blue lines. As can be seen, only up to 2 km of data is used for analyzing the ASD with a quite simple linear regression method.

Supplementary Fig. 26 Comparison between the standard method in previous publications^{8,9} and our new method for calculating the absent surface displacement (ASD). **a** The standard method, in which the displacement values outside “the damage zone” are assumed to be constant and only observations from within several hundred meters or several km from the fault are considered (Supplementary Table 1). **b** and **c** demonstrate our new method, where almost the entire coseismic displacement field as well as elastic model predictions are involved for analysis. It can be seen that the standard method underestimates the magnitude of the total slip, the ASD, and the width of the off-fault damage.

Milliner, C. W. D., Dolan, J. F., Hollingsworth, J., Leprince, S., & Ayoub, F. (2016). Comparison of coseismic near-field and off-fault surface deformation patterns of the 1992 Mw 7.3 Landers and 1999 Mw 7.1 Hector Mine earthquakes: Implications for controls on the distribution of surface strain. *Geophysical Research Letters*, 43(19), 10,115-10,124. <https://doi.org/10.1002/2016GL069841>

Scott, C. P., Arrowsmith, J. R., Nissen, E., Lajoie, L., Maruyama, T., & Chiba, T. (2018). The M7 2016 Kumamoto, Japan, Earthquake: 3-D Deformation Along the Fault and Within the Damage Zone Constrained From Differential Lidar Topography. *Journal of Geophysical Research: Solid Earth*, 123(7), 6138–6155. <https://doi.org/10.1029/2018JB015581>

23. Table S1 is a bit limited, and not very comprehensive.

We aim to list case studies about the off-fault damage based on the remote sensing displacement observations. We have carefully reviewed the literature and added more references, please let us know if we are missing any important studies.

Reviewer #2

Key results

The key message of this research is that the measured shallow coseismic offsets at the fault significantly underestimate the true near-surface deformation. The authors show that for the case of the 2023 Kahramanmaraş earthquakes the observed Absent Surface Deformation (ASD) was about 35% of the on-fault coseismic slip at depth. This result goes a long way towards addressing the shallow fault deficit, which has been a consistent observation for many earthquake ruptures around the world.

Validity

The presented results and conclusions are robust and follow a valid, clearly detailed methodology. The data and method as presented support the conclusions. However, I do have some relatively minor but important methodological points that I think need to be addressed. I detail these in a later section.

Significance

The result presented here is the first case where absent shallow deformation (ASD) has been estimated for a major earthquake. The authors show that ASD for the 2023 Kahramanmaraş earthquakes is on average 35% but could be up to 90% of the maximum modelled slip at depth. This is a significant observation that has important implications for the underestimation of long-term seismic hazard assessment based off earthquake offset measurements alone.

Data and methodology

The primary data used in this paper is openly available radar satellite data from the Sentinel-1 and ALOS-2 satellites. The authors present a robust processing methodology that combines routine time series analysis of long-term deformation with detailed and careful analysis of pseudo-3D deformation using a combination of pixel offsets, multiple aperture interferometry (MAI) and range spectrum split interferometry (RSSI).

Analytical approach

The principle analytical approach used in this work has been to use simple analytical models to estimate the expected coseismic slip at the fault. The authors fit the model to the observed displacement in the far field and determine the difference between the observed and modelled slip near the fault to estimate the absent surface deformation (ASD).

Thank you for your recommendation. Below, the comments are in **black**, our responses are in **red**, and the sentences modified in the manuscript are highlighted in **blue**. Figures supporting our perspective are labeled as **Figure R#** in the response letter.

Suggested improvements

I have three main points that I think could do with some additional work/clarity.

1. ASD ratios approaching 60% is very large! In some places this is equivalent to more than 5m of ‘missing shallow slip’. If all of this is attributed to off-fault deformation within ~5km either side of the fault, I would expect to see much more evidence of distributed slip. I realise the authors show examples of such slip in Figs 4e-d, but I would ask if the authors have attempted to sum up this distributed slip to see how much of the ASD it accounts for?

Thanks for your comments. We did not investigate the extent to which the ASD is accommodated by distributed slip. Simply put, the absent surface displacement (ASD) results primarily from two types of inelastic processes: secondary ruptures and plastic bulk deformation. To assess the magnitude of distributed slip (i.e., the plastic bulk deformation), we should exclude the offsets induced by the secondary ruptures. However, since secondary ruptures are generally only a few hundred meters or less in length, it is difficult for our SAR-based displacement data, with a 50-meter resolution, to identify these features. The L-band ALOS-2 interferograms shown in Figs. 4d-f have the potential to distinguish between distributed slip and secondary ruptures, but accurately unwrapping the interferograms in near-fault regions is challenging. Therefore, based on the available datasets, we can only qualitatively correlate the interferogram features to the off-fault damage rather than quantitatively assess the contribution of secondary ruptures and plastic bulk deformation.

We added the following sentences to clarify this point in the revised manuscript (Lines 252-255):

“Unwrapped ALOS-2 interferograms would allow for differentiating between secondary fault offsets and inelastic deformation within observed ASD areas. However, unwrapping interferograms near coseismic fault ruptures of large earthquakes is usually challenging.”

2. I think the authors haven't given sufficient attention to the role of shallow afterslip in accommodating some of the missing coseismic ASD. In other well documented earthquakes in Turkiye (e.g. Izmit 1999) we see that shallow afterslip rates can be very significant. For example, Hussain et al., (2016) showed that shallow afterslip probably contributed over 1m of additional slip in the shallow portions of the fault. While this might not detract from the conclusions of this paper, it might reduce the overall ASD fraction. I realise that the authors touch on this in Fig 4a (and associated main text) but I don't think this is sufficient. Perhaps one way to investigate the shallow afterslip in detail is by differencing the displacement time series of pixels close to the fault/near the fault within the ASD width.

Thanks for your suggestion. We investigated the displacement difference across the fault for the interseismic and postseismic periods. The following Supplementary Fig. 18 suggests that although shallow cm-level postseismic slip can be observed along the northeastern part of the main ruptures, it is negligible compared to the meter-level coseismic absent surface displacement (ASD). To better illuminate this point, Supplementary Fig. 18 has been added in the revised supplementary information and mentioned in Line 235 of the revised main manuscript.

“Indeed, shallow afterslip only exists in specific locations (e.g., at the northeastern end of the first rupture⁴⁶) and the amplitude does not exceed few tens of centimeters (Supplementary Fig. 18), far less than is needed to catch up with the modeled meter-scale SSD.”

Supplementary Fig. 18 The displacement difference across the fault for the interseismic and postseismic periods. **a** and **b** are interseismic and postseismic east-west displacement rates, respectively, where the blue lines represent the main coseismic ruptures and the labeled numbers are the distance in km along the rupture from the southwest. In **b**, the black markers A-A' and B-B' at the northeastern ends of the ruptures are selected point pairs to show the difference of displacement time series across the rupture. Point pair A and A' (and B and B') are two points located on opposite sides and 1 km away from the rupture. **c** The difference of interseismic displacement rate across the fault along the two main ruptures, calculated by differencing the displacement rate value within 1.5 km range on each side of the rupture. **d** is same as **c** but for the postseismic displacement rate. **e** The difference of postseismic displacement time series at point pairs A-A' and B-B' for the ascending T116 Sentinel-1 dataset. **f** is same as **e** but for the descending T21 Sentinel-1 dataset. In **c**, there is negligible difference of interseismic displacement rate between the two sides of the near-fault regions while **d** suggests 0-3 cm/yr shallow postseismic slip along most parts of the main ruptures, except at the northeastern end of the two ruptures, where it is 5-10 cm/yr. Although there is also an obvious contrast in displacement rate across the rupture at the southwestern end of the main rupture of Mw7.8 event, it mainly results from the 20th Feb. 2023 Mw6.3 aftershock. In **e** and **f**, we show the difference of postseismic displacement time series at the two selected point pairs A-A' and B-B', indicating a decaying shallow slip pattern around the eastern end of the main ruptures. This cm-level shallow slip can compensate only a small part of the missing meter-level coseismic absent surface displacement.

3. A fundamental and well-known challenge with the interseismic and postseismic observations on at least the southern segment of the Mw 7.8 rupture is that this fault is not optimally oriented for InSAR deformation measurements. Can the authors be confident that they have captured the full long term and early postseismic deformation here?

Thanks for your comment. To investigate whether there is shallow postseismic deformation around the southern segment of the Mw 7.8 rupture, we employed a burst-overlap InSAR (BOI) method to derive the azimuth displacement (near north-south direction) over the burst overlap areas of Sentinel-1 SAR data. The following Supplementary Fig. 17 shows that there is negligible north-south postseismic shallow slip close to the southern segment of the Mw 7.8 rupture. We have not processed the interseismic BOI displacement around the southern

segment due to the large volume of SAR images from 2014-2023 as well as the low signal-to-noise ratio of the interseismic displacement. Therefore, based on what geodetic observations we have, there appears to be negligible interseismic and postseismic deformation to compensate the observed coseismic absent surface displacement (ASD). In the revised supplementary information, we added the newly processed BOI result in Supplementary Fig. 17. Besides, in the main manuscript, the following sentences declare our concern on how representative our short-term geodetic observations are for the whole earthquake cycle.

“The moment of the estimated SSD needed to compensate for the observed ASD is 5.65×10^{19} Nm, corresponding to a Mw 7.1 event (Supplementary Fig. 16). However, SAR-based interseismic deformation mapping⁴³, based on data from 2014-2019, indicates no surface creep on the fault sections activated in the Kahramanmaraş earthquakes (Fig. 4a), which rules out shallow fault slip in the years before the earthquakes and suggests that creep is not common along the ruptured sections of the EAF in general. Even if transient slow-slip events did occur before the InSAR observation period, their magnitude would typically be only a few millimeters, with recurrence intervals of several years^{44,45}. It is thus unlikely that transient slow-slip events compensated for the SSD of 2-4 meters during historical times without being detected.” (Lines 220-229)

“Therefore, while we cannot exclude the possibility of shallow slow-slip events before 2014, the current interseismic and postseismic data do not support that reduced shallow elastic slip is responsible for observed ASD.” (Lines 238-240)

Supplementary Fig. 17 Average postseismic displacement rates for the first 9 months after the earthquakes, based postseismic Sentinel-1 SAR images. **a**, **b**, and **c** are the InSAR line-of-sight displacement rates for ascending tracks 14 and 116, and descending track 21, respectively. **d**, **e**, and **f** are the corresponding burst-overlap InSAR (BOI) azimuth displacement rates. The time span of the Sentinel-1 SAR images used here is 20230209-20231031, 20230228-20231107, and 20230210-20231101 for the three tracks, using a date format of `yyyymmdd`. Magenta lines mark the fault surface ruptures, and the black squares in a-c indicate the reference location for each track. It can be seen that there is negligible postseismic shallow slip around most segments of the main ruptures except for the northeastern end of ruptures. However, this postseismic shallow slip around the northeastern end is far smaller than needed to catch up with the observed meter-scale absent surface displacement (ASD), as demonstrated in Supplementary Fig. 18. Since the spatial extent of the postseismic displacements is notably larger than that of the coseismic displacement, most of the postseismic displacements are likely due to deep viscoelastic processes.

Additionally, some minor points are listed below:

1. Lines: 122-123: Figure labelling incorrect. I think you meant Fig. 2d. In general, some of the figures are labelled incorrectly and/or incorrectly referenced in the main text. Please check these.

Thanks for your comments, we have carefully checked the labels and references in the manuscript.

2. Line 268: Slip rate estimates of... [what?]

We removed “of” in the revised manuscript.

3. The discussion presented in between Lines 243 and 260 feels incomplete. There have been plenty of detailed studies of coseismic earthquake offsets on larger spatial scales with both radar and optical datasets (e.g. Napa Valley, Ridgecrest, Kaikoura, Palu). Perhaps a better comparison could be made with these studies in addition to the subset of studies that have used optical images of near-fault deformation.

We modified the original sentences as the follows and added some related references (Lines 283-286)

“This limitation prevents them from precisely deciphering the total far-field elastic deformation curve across the fault (i.e., the red model curve in Fig. 2d), even though the optical images are able to reveal the entire width of deformation pattern of earthquakes^{50-52.}”

50. Jolivet R, et al. *The 2013 Mw 7.7 Balochistan Earthquake: Seismic Potential of an Accretionary Wedge. Bull. Seismol. Soc. Am.* 104, 1020-1030 (2014).

51. Provost F, et al. *High-resolution co-seismic fault offsets of the 2023 Türkiye earthquake ruptures using satellite imagery. Sci. Rep.* 14, 6834 (2024).

52. He LJ, et al. *Surface Displacement and Source Model Separation of the Two Strongest Earthquakes During the 2019 Ridgecrest Sequence: Insights From InSAR, GPS, and Optical Data. Journal of Geophysical Research-Solid Earth* 127, e2021JB022779 (2022).

Clarity and context

The authors provide a succinct summary of the problem and explain the context of their findings clearly and concisely. It would have been good to read a little more about the broader implications of their work alongside a discussion of potential limitations (for example due to the viewing geometry issues suggested above).

See the above response to the main comment #3.

References

The references are sufficient and appropriate. Although the authors may choose to include additional citations to cover the suggested additions above.

We have added more references in the revised manuscript.

Reviewer #3

Thanks very much to the authors for a compelling and well-written manuscript. This study takes advantage of both Sentinel-1 and ALOS-2 SAR data to calculate the coseismic displacements using dInSAR, MAI, POT, and RSSI methods, and then estimate the 3D coseismic displacements using the SM-VCE approach, for the 2023 Kahramanmaras M7.8 and M7.6 earthquakes. From the 3D displacements, the authors estimate a metric they call the “absent surface displacement” (ASD), which they estimate all along both fault ruptures. The data indicate that ASD can vary along strike and appears to correlate strongly with fault complexity (e.g., complex bends or stepovers result in higher ASD while simple fault stretches have lower ASD), and that, on average, ~35% of fault slip at depth can get “eaten up” by ASD/inelastic deformation near or at the surface. In addition, the data indicate that looking further afield may be very important to determine an accurate measure of ASD/inelastic fault deformation, and that previous studies likely have not looked far-enough away from the rupture to capture the full measurement of ASD. All of this suggests that previous studies have likely been underestimating fault slip (both by not including the full aperture of ASD/off-fault deformation width, and by focusing only on slip measurements along the fault rupture plane), and therefore potentially underestimating fault hazards.

This was a fascinating paper, and I support publication as I believe it will be an important catalyst for continued discussions about how we even just *measure* how an earthquake has deformed the surface, let alone how that flows into our estimates of earthquake hazards along known fault systems. I have a couple line-by-line comments below, but they are mainly cosmetic suggestions/typo corrections. I think the authors have done their due diligence to provide all of their work and their supporting data for this effort.

Thanks for your recommendation. Below, the comments are in **black**, our responses are in **red**, and the sentences modified in the manuscript are highlighted in **blue**. Figures supporting our perspective are labeled as **Figure R#** in the response letter.

Line-by-line comments:

L19: Maybe should be “experienced higher levels of off-fault damage” or “experienced a higher level of...”

L40: I think there may be an article missing, perhaps should be “embedded in “an” homogenous or layered elastic...”

Figure 1: Nice figure, very clear

Figure 2: What are the uncertainties on your profile measurement? (and from that, what are the uncertainties on the estimate of ASD?) In (d), should it be labeled fault-parallel displacement to be specific within the figure? (I see it says it in the caption, but it might be beneficial to label it on the axis too)

We have corrected the typos. As for the uncertainty of the profile measurement, it should be the level of the uncertainty of east-west and north-south displacements since the displacement at each point on the profile is obtained by averaging the displacement values of the nearest four points. It is difficult to determine the uncertainty of east-west and north-south displacements for all SAR points. Eight GNSS stations within the study area were used to quantify the overall accuracy of our SAR-based displacements, yielding a root mean square error of the difference between our SAR-based 3D displacements and GNSS observations of 5.5 cm, 8.6 cm, and 5.8 cm for the east, north, and vertical displacement components, respectively. In this case, we

would say the uncertainty of the profile displacement is 5-9 cm. We added the following sentences in the figure caption to clarify this point (Lines 99-101):

“The uncertainty of the profile displacements is 5-9 cm, as indicated by the comparison of the SAR-based 3D displacements with GNSS observations (Supplementary Fig. 9).”

Question: How trustworthy are the nearest fault measurements? They seem to be critical in order to make this estimate of ASD, but are there any issues with loss of coherence near the fault trace?

We validated the near-fault measurements in two different ways, in response to the first two comments of Reviewer #1. (1) Independent Sentinel-2 optical displacement derivations were used to validate our SAR-based horizontal displacements, and (2) we reprojected the 3D displacements to the geometry of SAR image offset observations to see how well the 3D displacements fit with the SAR observations. The results provide a validation of our SAR-based 3D displacement in the near-fault regions and have been added to the main manuscript and supplementary information, see replies to the first two comments of Reviewer #1.

As for the loss of coherence in the near-fault regions, it is the case for the phase-based measurements (e.g., DInSAR and MAI), but for the amplitude-based measurements (i.e., POT), it is feasible to obtain sufficient observations around the near-fault regions, see replies to comment no. 2 by Reviewer #1 and new Supplementary Figs. 7-8.

Figure 3: Inside the subset, I am a little confused as to why the binned ratios are plotted as being the same on both side of the fault, whereas in the main 3a figure, the ratios appear pretty different in a couple places (on both sides of the fault – e.g., the NE-most end of the main M7.8 rupture). Are you just plotting the highest percentage from either sides? Maybe I am missing something.

We didn't consider the lateral width of the damage zone in this insert map. What we want to emphasize here is segmentation of the ratio of absent surface displacement (ASD). The wide colorful zone here is just for better visualizing the segmentation. To clarify this point, we added the following sentence in the Fig. 3 caption in the revised manuscript.

“The classes of ASDr values along the ruptures are plotted as thick lines to better visualize segmentations.”

L164: just add an article again, for “with the maximum slip at “a” depth of ~5km”.

Thanks, we have corrected this typo.

Figure 4: In (c), what do the triangles at 50 km from the EAF represent? Are they marking something specific or are they just label pointers? For (e),(f), and (d), can you put the non-annotated versions of the wrapped interferograms, so the reader can look for and see the discontinuities for themselves?

The triangles are just label pointers. To prevent misunderstanding, modifications have been made to the label pointers. The clear version of d-f is shown in the revised Supplementary Fig. 15.

Supplementary Fig. 15 Same as Figs. 4d-f in the main text but without text, arrows, and fringe discontinuity annotations.

Just a thought: have you considered calculating the phase gradient maps for these? I enjoyed the maps produced by Xiaohua Xu of the 2019 Ridgecrest, CA sequence (Xu et al., 2020, Science), and they seem to highlight this type of off-fault deformation very clearly

We didn't calculate the phase gradient maps from InSAR interferograms since we focus on the near-fault regions where the fringes are too dense or distorted to obtain a good gradient map. However, we calculated the displacement gradient for the fault-parallel displacement along the fault-perpendicular profiles. As shown in the following Figure R3, the extent of the positive displacement gradient (blue color) is well consistent with the obtained damage zone (black lines).

[panel redacted]

Figure R3. Left shows displacement gradients of fault-parallel displacements on fault-perpendicular profiles. For a profile, the general shape of the displacement looks like in the right panel (same as Fig. 2d in the main text), where the displacement gradient is calculated from left to right and the gradient near the fault should contain negative values in case of perfectly elastic deformation, but shows positive values within the damage zone (blue color in left).

L201-205: I don't necessarily disagree, but four years of InSAR measurements can't tell you that this fault hasn't crept during its "lifetime", can it? Are there any papers that confirm this observation? How many "creep events" would you need to accommodate this magnitude of deformation?

We agree with you that we can't rule out that possible creep events occurred outside the InSAR observation period. However, from research on typical strike-slip continental faults (e.g., the North Anatolian Fault and the San Andreas Fault), the shallow transient slip is typically only a few millimeters, with recurrence intervals of several years^{44,45}. Even if we assume that the average transient slip is 10 mm and the average SSD is 2 m, two hundred creep events would be required to compensate for the SSD. To better explain these points, we added the following sentences to the revised manuscript. (Lines 226-229, Lines 238-240)

"Even if transient slow-slip events did occur before the InSAR observation period, their magnitude would typically be only a few millimeters, with recurrence intervals of several years^{44,45}. It is thus unlikely that transient slow-slip events compensated for the SSD of 2-4 meters during historical times without being detected."

"Therefore, while we cannot exclude the possibility of shallow slow-slip events before 2014, the current interseismic and postseismic data do not support that reduced shallow elastic slip is responsible for observed ASD."

44. Rousset B, et al. An aseismic slip transient on the North Anatolian Fault. *Geophys. Res. Lett.* 43, 3254-3262 (2016).

45. Tymofyeyeva E, et al. Slow Slip Event On the Southern San Andreas Fault Triggered by the 2017 Mw8.2 Chiapas (Mexico) Earthquake. *Journal of Geophysical Research-Solid Earth* 124, 9956-9975 (2019).

L212: Add an "a" in "does not exceed "a" few tens of centimeters"

Thanks, we have corrected this.

L214: At a postseismic rate of 20 cm/yr, in one year, it would make up 1 meter of displacement – are you suggesting most of this is viscoelastic, and therefore not occurring on the shallow fault section?

Since the spatial extent of the postseismic displacement is much larger than that of the coseismic displacement, we prefer to attribute most of the postseismic displacement to deep viscoelastic processes. To better clarify this point, we have modified the caption of Supplementary Fig. 17 (the postseismic InSAR displacement rates) in the revised supplementary information. See reply to comment 3 made by Reviewer #2.

L268: "...slip rate estimates of..." of what? Maybe missing a word? Or maybe just delete "of"?

Thanks, we have corrected this.

Reviewer #1

I've gone through the revisions, and the authors have addressed my previous comments.

Thank you for your time of reviewing our manuscript.

Reviewer #2

I very much enjoyed reading this manuscript again. I would like to thank the authors for the level of detail and care they have given in responding to the reviews. The paper makes a significant and valuable contribution to the body of knowledge around coseismic deformation particularly in quantifying the near-fault deformation. I'm happy to recommend the paper for publication.

Thank you for your recommendation.

Reviewer #3

My questions and concerns have been answered and addressed, and I still support publication. I think the authors have done a very thorough job responding to my and the other reviewers' comments and questions, and I thank them for their clear and detailed responses.

Thank you for your support and recommendation.